# Telophase correction refines division orientation in stratified epithelia

Kendall J Lough[1,2], Kevin M Byrd[1,2,3], Carlos P Descovich[1,2], Danielle C Spitzer[1,2†], Abby J Bergman[1,2], Gerard MJ Beaudoin III[4,5‡], Louis F Reichardt[4,5], Scott E Williams[1,2]*

[1]Department of Pathology and Laboratory Medicine, Lineberger Comprehensive Cancer Center, The University of North Carolina, Chapel Hill, United States; [2]Department of Biology, Lineberger Comprehensive Cancer Centre, The University of North Carolina, Chapel Hill, United States; [3]Department of Oral & Craniofacial Health Sciences, The University of North Carolina School of Dentistry, Chapel Hill, United States; [4]Department of Biochemistry & Biophysics, University of California, San Francisco, San Francisco, United States; [5]Department of Physiology, University of California, San Francisco, San Francisco, United States

*For correspondence:
scott_williams@med.unc.edu

Present address: †Molecular and Cellular Biology Program, University of California, Berkeley, United States; ‡Department of Biology, Trinity University, San Antonio, United States

Competing interests: The authors declare that no competing interests exist.

**Abstract** During organogenesis, precise control of spindle orientation balances proliferation and differentiation. In the developing murine epidermis, planar and perpendicular divisions yield symmetric and asymmetric fate outcomes, respectively. Classically, division axis specification involves centrosome migration and spindle rotation, events occurring early in mitosis. Here, we identify a novel orientation mechanism which corrects erroneous anaphase orientations during telophase. The directionality of reorientation correlates with the maintenance or loss of basal contact by the apical daughter. While the scaffolding protein LGN is known to determine initial spindle positioning, we show that LGN also functions during telophase to reorient oblique divisions toward perpendicular. The fidelity of telophase correction also relies on the tension-sensitive adherens junction proteins vinculin, α-E-catenin, and afadin. Failure of this corrective mechanism impacts tissue architecture, as persistent oblique divisions induce precocious, sustained differentiation. The division orientation plasticity provided by telophase correction may enable progenitors to adapt to local tissue needs.

## Introduction

Stem and progenitor cells utilize asymmetric cell divisions to balance self-renewal and differentiation. Cell fate decisions can be influenced by the division axis, with the choice between symmetric and asymmetric fate outcomes dictated by positioning of the mitotic spindle. Mechanistically, precise control of division orientation may serve to equally or unequally partition fate determinants, or restrict access to a stem cell niche (*Knoblich, 2008*; *Siller and Doe, 2009*). Errors in division orientation can lead to defects in differentiation and cell identity, with the potential to drive overgrowths associated with cancer (*Knoblich, 2010*; *Martin-Belmonte and Perez-Moreno, 2011*; *Neumüller and Knoblich, 2009*).

The developing murine epidermis serves as an excellent model for studying how oriented cell divisions direct cell fate choices. Basal progenitors are capable of dividing either within the plane of the epithelium or perpendicular to it, resulting in symmetric or asymmetric divisions, respectively (*Lechler and Fuchs, 2005*; *Smart, 1970*). This process is governed by a conserved complex of spindle orienting proteins, including the essential linker LGN/Gpsm2 (*Williams et al., 2011*; *Williams et al., 2014*). During epidermal and oral epithelial stratification, LGN is recruited to the

apical cortex in ~50% of mitoses, and LGN loss leads to increased planar divisions and severe differentiation defects (*Byrd et al., 2016*; *Williams et al., 2011*; *Williams et al., 2014*). Thus, a parsimonious explanation for the observed bimodal distribution of division angles is that perpendicular divisions occur when sufficient levels of LGN are recruited to the apical cortex during early mitosis, and planar divisions occur when this apical recruitment fails.

In this and other models, it is assumed that the division axis is established relatively early in mitosis, either through directed centrosome migration or spindle rotation. As an example of the former, in the *Drosophila melanogaster* testis and larval neuroblasts, one centrosome migrates to the opposite side of the cell during prophase, and the metaphase spindle forms along, and remains fixed by, this centrosomal axis (*Rebollo et al., 2009*; *Siller et al., 2006*; *Yamashita et al., 2003*). In other systems—including the *C. elegans* early embryo, *D. melanogaster* embryonic neuroblasts, and progenitors of the vertebrate neuroepithelia—the spindle dynamically rotates during metaphase to align with extrinsic niche-derived or intrinsic polarity cues (*Geldmacher-Voss et al., 2003*; *Haydar et al., 2003*; *Hyman and White, 1987*; *Kaltschmidt et al., 2000*). Collectively, these studies support the view that spindle orientation generally operates prior to anaphase onset.

On the other hand, there are hints from other studies that the metaphase-anaphase transition involves dynamic reorganization of the spindle orientation machinery. For example, in HeLa cells it has been shown that while LGN is essential for NuMA localization during early mitosis, LGN becomes dispensable during anaphase, when NuMA's cortical localization is dependent upon phosphoinositides (*Kotak et al., 2014*). However, whether LGN functions to orient spindles at late stages of mitosis in other, polarized cell types, remains unknown.

Here, utilizing ex vivo live imaging in combination with mosaic RNAi, we find that division orientation in the developing murine epidermis is not determined solely by LGN localization during early mitosis. Surprisingly, LGN appears to play a "maintenance" role during anaphase/telophase, while an LGN-independent pathway involving adherens junction (AJ) proteins also acts to refine imprecise initial spindle positioning. We show that spindle orientation remains dynamic even into late stages of mitosis, and surprisingly, division axes remain random and uncommitted long after metaphase. While most cells enter anaphase with planar (0–30°) or perpendicular (60–90°) orientations and maintain this division axis through telophase, a significant proportion (30–40%) are initially oriented obliquely (30–60°), but undergo dramatic reorientation, a process we term telophase correction. In addition, we demonstrate that the α-E-catenin/vinculin/afadin cytoskeletal scaffolding complex is required for this correction to occur, and likely functions to modulate the tensile properties of the cell cortex by altering how actin is recruited to AJs. Mutants defective for telophase correction display precocious stratification which persists into later stages, highlighting the importance for this mechanism in generating normal tissue architecture. Furthermore, using genetic lineage tracing in *afadin* (*Afdn*) mutants, we confirm that uncorrected oblique divisions result in a strong bias toward differentiation over self-renewal.

Collectively, these studies support a novel two-step model of oriented cell division, where intrinsic factors such as LGN provide spatial cues that guide initial spindle positioning during early mitosis, while extrinsic factors such as cell-cell adhesions may provide a tension or density-sensing mechanism that refines the division plane during telophase to ensure normal tissue architecture. Our data further suggest that these mechanisms are modulated over developmental time to coordinate progenitor-expansive and differentiative programs.

## Results

### Randomized division orientation persists into anaphase

During peak stratification, epidermal basal cells undergo either LGN-dependent perpendicular divisions or LGN-independent planar divisions, with roughly equal frequency. LGN is invariably apical when recruited to the cell cortex during prophase and remains apical at telophase in perpendicular divisions (*Williams et al., 2011*; *Williams et al., 2014*). However, this bimodal distribution of division angles only emerges by ~E16.5, because the apical polarization of LGN is less efficient at earlier ages, resulting in a high proportion of oblique angles, and fewer perpendicular divisions (*Williams et al., 2014*). Our previous studies reported a bimodal distribution of division angles at late stages of mitosis and randomized division angles during metaphase (*Williams et al., 2011*;

*Williams et al., 2014*), while other groups have reported that spindle rotation occurs during prometaphase and is fixed to a bimodal distribution by late metaphase/early anaphase (*Poulson and Lechler, 2010*; *Seldin et al., 2016*). While these studies agree that spindle rotation occurs, they come to different conclusions about when and how the spindle axis becomes fixed to a bimodal distribution.

Because these studies vary in the ages examined and the method used to identify mitotic cells at specific stages, we sought to apply a rigorous and unambiguous methodology to identify metaphase, anaphase and telophase cells at a single timepoint (E16.5), when nearly all divisions are either planar or perpendicular. Because phosphorylation at Ser10 and Ser28 of histone-H3 (pHH3) declines rapidly after metaphase (*Hans and Dimitrov, 2001*) antibodies raised against pHH3 vary in their ability to detect anaphase cells, and do not label telophase cells at all. Thus, we used another marker, Survivin (Birc5), which localizes to centromeres through metaphase, and redistributes to central spindle fibers and the cleavage furrow during anaphase and telophase, respectively (*Beardmore et al., 2004*; *Caldas et al., 2005*). In this manner, anaphase and telophase cells can be readily distinguished by their pattern of Survivin staining (*Figure 1A*). Since its original use (*Williams et al., 2011*), Survivin has been used by multiple groups across a variety of tissues to measure the division axis (*Aragona et al., 2017*; *Asrani et al., 2017*; *Byrd et al., 2016*; *Cohen et al., 2019*; *Ding et al., 2016*; *Dor-On et al., 2017*; *Ellis et al., 2019*; *Ichijo et al., 2017*; *Jones et al., 2019*; *Liu et al., 2019*; *Niessen et al., 2013*; *Wang et al., 2018*; *Williams et al., 2014*).

We examined a large cohort of fixed sections of dorsal back skin epidermis from twenty-five E16.5 mouse embryos of varying strains (CD1, 129S4/SvJae and C57Bl6/J), and identified and imaged 536 Survivin+ metaphase, anaphase and telophase cells. We noted that anaphase and metaphase cells were comparatively rare, each occurring at ~1/5 the frequency of telophase cells (*Figure 1B*). In agreement with our previous observation, metaphase plates were oriented randomly, suggesting that spindle rotation occurs during metaphase. Surprisingly, however, the distribution of division angles remained random at anaphase, only establishing a bimodal distribution during telophase (*Figure 1B,C*). This trend held for each mouse strain (*Figure 1—figure supplement 1A*), demonstrating that differences in genetic background are unlikely to explain discrepancies in anaphase orientation reported by our group and others. Of note, because of the relative scarcity of anaphase divisions, they make a negligible contribution (*Figure 1B*, compare 'telophase" to "ana+telo"), perhaps explaining why in previous studies, so few oblique divisions are reported in total Survivin+ pools. Nonetheless, these data demonstrate that basal cells remain uncommitted to a final plane of division beyond metaphase, and suggest that a previously uncharacterized spindle orientation mechanism occurs after anaphase onset.

## Oblique anaphase divisions reorient during telophase

As a next step, we performed ex vivo live imaging of E16.5 embryonic epidermal explants (*Cetera et al., 2018*), in order to examine the dynamics of spindle orientation at late stages of mitosis. To easily discriminate the basal layer of the epidermis from the underlying dermis and visualize cell nuclei during mitosis we utilized two combinations of alleles: 1) $Rosa26^{mT/mG}$ + $Krt14^{Cre}$, where cell membranes are GFP+ in the epidermis and tdTomato+ in the dermis, and 2) $Rosa26^{mT/mG}$ + $Krt14^{H2B-GFP}$, where epidermal cell membranes are tdTomato+ and nuclei are H2B-GFP+ (*Figure 1D*). In both allele combinations, accurate measurements of the division angle relative to the epidermal-dermal border could be made in z-projections (*Figure 1E*). The $Rosa26^{mT/mG}$ + $Krt14^{H2B-GFP}$ combination was particularly useful for visualizing both the initiation of cleavage furrow ingression and the separation of nuclei that occurs at anaphase onset (defined as t = 0). Since cell nuclei could not be visualized in the $Rosa26^{mT/mG}$ + $Krt14^{Cre}$ background, we defined anaphase onset as the frame in which cleavage furrow ingression could be first visualized. Of note, in both allele combinations, the duration of anaphase was observed to be short—typically two 5' frames elapsed where mGFP or mtdTom was not visible between daughter nuclei—providing an explanation for why anaphase cells were rarely observed in fixed tissue.

In both imaging paradigms, we observed a high proportion (~2/3) of basal cells which entered anaphase at either planar or perpendicular orientations that remained relatively fixed for the duration of the imaging period (*Figure 1—figure supplement 1B, C*; *Figure 1—videos 1, 2*). However, as suggested by our analyses of fixed tissue, many basal progenitors also frequently initiated anaphase at oblique angles (*Figure 1E*; t=0, φ=division angle). Notably, these oblique divisions invariably corrected to either planar or perpendicular within an hour (*Figure 1E*; *Figure 1—figure*



**Figure 1.** Telophase reorientation corrects oblique anaphase orientations. (**A**) Sagittal sections from E16.5 embryos showing mitotic basal cells at indicated stages. Yellow arrows indicate division axis relative to basement membrane (dashed white line). Apical LGN (red) is generally present in oblique and perpendicular divisions, but absent from planar divisions. Survivin (green) is diffusely distributed between daughter pairs at anaphase, transitioning to stereotypic dual-puncta by telophase. (**B**) Radial histograms of division orientation at metaphase, anaphase, telophase and anaphase +telophase in E16.5 wild-type controls; *n* indicates number of divisions measured from >20 embryos per mitotic stage. (**C**) Same data as in (**B**), plotted as a cumulative frequency distribution. Note sigmoidal pattern at telophase (black, solid line), characteristic of bimodal distribution of division angles. Compare to linear pattern, characteristic of random distributions at metaphase (red) and anaphase (blue). (**D**) Schematic of experimental design for live imaging of embryonic epidermal explants. *Krt14^{Cre}*; *Rosa26^{mT/mG}* is used to label epidermis with membrane (m)-GFP and other tissues (including dermis) with mTdTomato. Alternatively, *Krt14^{H2B-GFP}* is used to label nuclei while *Rosa26^{mT/mG}* without Cre ubiquitously labels cells with membrane-tdTomato. (**E**) Z-projection stills from a movie of a *Krt14^{Cre}*; *Rosa26^{mT/mG}* (top) and *Krt14^{H2B-GFP}*; *Rosa26^{mT/mG}* (bottom) mitotic cell as it enters anaphase (defined as t = 0), through 60 min post-anaphase onset, depicting planar telophase correction. Epidermal-dermal boundary shown by red line. Dividing daughter pairs are outlined with yellow dashed lines. Division orientation angles are shown below (φ, anaphase onset; θ, +1 hr). (**F**) Traces of division orientation at five minute intervals for 15 representative cells from telophase onset to +1 hr. (**G**) Cumulative frequency distribution of division angles from *Krt14^{Cre}*; *Rosa26^{mT/mG}* live imaging experiments of E16.5 embryos at anpahse onset (blue; φ) and +1 hr later (black; θ). *n* indicates number of divisions from 4 embryos across four independent sessions. (**H**) Data from (**G**) depicting division orientations at anaphase onset and 1 hr later. Connecting lines demonstrate that ~60% oblique anaphase divisions reorient to planar (black lines) while the remaining ~40% correct to perpendicular (gray lines). Scale bars, 5 µm (**A**), 10 µm (**E**). **p<0.01, ***p<0.001, by Kolmogorov-Smirnov test. See also *Figure 1—figure supplement 1*.

The online version of this article includes the following video, source data, and figure supplement(s) for figure 1:

**Source data 1.** Original measurements used to generate panels B, C, F, G, H.

**Figure supplement 1.** Telophase reorientation corrects oblique anaphase orientations.

*Figure 1 continued on next page*

*Figure 1 continued*

**Figure 1—video 1.** Planar anaphase orientation is fixed.
https://elifesciences.org/articles/49249#fig1video1
**Figure 1—video 2.** Perpendicular anaphase orientation is fixed.
https://elifesciences.org/articles/49249#fig1video2
**Figure 1—video 3.** Oblique anaphase orientations undergo planar telophase reorientation.
https://elifesciences.org/articles/49249#fig1video3
**Figure 1—video 4.** Oblique anaphase divisions display perpendicular correction.
https://elifesciences.org/articles/49249#fig1video4

*supplement 1D, E*; *Figure 1—videos 3*, *4*). When the angle of division was plotted over time, we noted that this reorientation, hereafter referred to as telophase correction, generally occurred within the first 30 minutes of anaphase onset. (*Figure 1F*). Since little or no reorientation occurred after 1h, we assigned this as the imaging endpoint (t=+60min, θ=division angle). Of note, the distribution of division angles observed in these movies at anaphase onset (φ) and 1h later (θ) was remarkably similar to the distribution of anaphase and telophase orientations observed in fixed tissue (compare *Figure 1G to 1C*). When the behavior of individual cells was plotted at anaphase onset relative to 1h later, we observed that when φ>60°, correction tended to occur toward perpendicular, and when φ<30°, correction tended to occur toward planar, while oblique angles were less predictable (*Figure 1H*). This suggested that the directionality of correction is not purely stochastic.

## LGN mediates perpendicular telophase correction

Previous studies have shown that LGN (Pins in *Drosophila*)—along with its binding partners Insc (Inscuteable), NuMA (Mud), and Gαi—play key roles in oriented cell divisions (*Bowman et al., 2006*; *Du and Macara, 2004*; *Izumi et al., 2006*; *Kraut et al., 1996*; *Mora-Bermúdez et al., 2014*; *Schaefer et al., 2000*; *Siller et al., 2006*; *Williams et al., 2014*; *Zigman et al., 2005*). In the conventional view, LGN functions primarily during prometaphase-metaphase by facilitating capture and anchoring of astral microtubules to the cell cortex. In developing stratified epithelia, LGN first localizes to the apical cortex during prophase (*Byrd et al., 2016*; *Lechler and Fuchs, 2005*; *Williams et al., 2011*; *Williams et al., 2014*). However, our finding that a large proportion of anaphase cells are oriented obliquely suggests that initial perpendicular spindle positioning by LGN may be imprecise, and raises the question of whether LGN may also function during perpendicular telophase correction.

To test this, we performed ex vivo live imaging of $Krt14^{Cre}$; $Rosa26^{mT/mG}$ epidermal explants mosaically-transduced with a previously validated shRNA targeting LGN/Gpsm2 ($Gpsm2^{1617}$) or non-targeting Scramble shRNA control (*Williams et al., 2011*; *Williams et al., 2014*). The H2B-RFP reporter allowed us to track pronuclear separation during anaphase onset, and distinguish RFP+ transduced/knockdown basal cells from non-transduced/wild-type RFP- internal controls (*Figure 2A, B*). Like wild-type explants, Scramble RFP+ and $Gpsm2^{1617}$ RFP- control cells were randomly oriented at anaphase onset, but corrected to a bimodal distribution 1h later (*Figure 2C–E*). Compared to wild-type cells, a higher proportion of $Gpsm2^{1617}$ RFP+ cells entered anaphase at planar (φ <30) orientations (75% vs 30% for $Gpsm2^{1617}$ RFP- and 32% for Scramble RFP+). In addition, very few $Gpsm2^{1617}$ RFP+ cells (2%, n = 49) entered anaphase at perpendicular (φ>60) orientations (*Figure 2D,E*). These data support our previous findings that LGN is required for initial positioning of perpendicular spindles. Interestingly, however, the minority (23%) of $Gpsm2^{1617}$ RFP+ cells that entered anaphase at oblique angles invariably corrected toward planar (*Figure 2D,E*). Taken together, these data suggest that, in addition to its known role in orienting spindles along the apicobasal axis during prometaphase, LGN also serves a second maintenance function later in mitosis, where it promotes perpendicular correction during telophase.

## Directionality of telophase correction is correlated with basement membrane contact

We next sought to address the mechanisms underlying planar directed telophase correction. In our wild-type live imaging experiments, we observed that while initial orientations of φ >60° typically corrected to perpendicular, and φ <30° to planar, the behavior of intermediate orientations (φ = 30–60°)

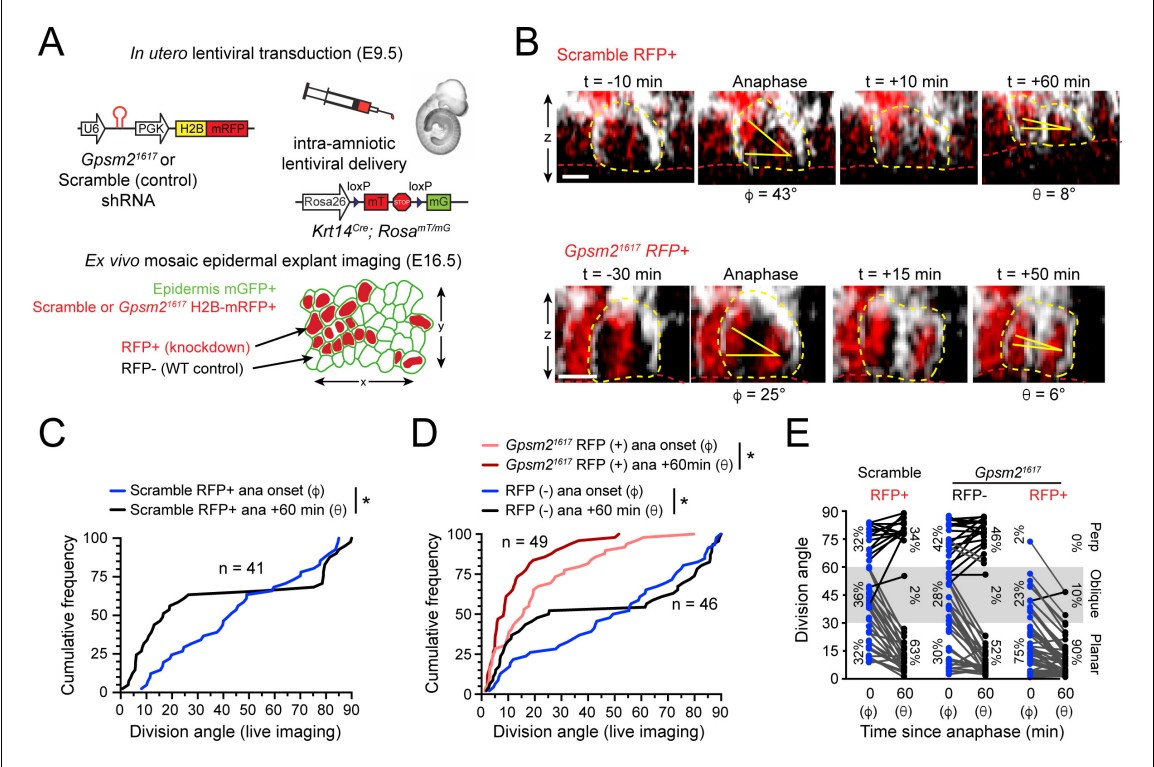

**Figure 2.** LGN mediates perpendicular but not planar telophase correction. (**A**) Schematic of modified experimental protocol of live imaging of epidermal explants (see **Figure 1D**) incorporating lentiviral shRNA transduction to generate mosaic knockdown tissue. Transduced/knockdown regions are marked with histone H2B-mRFP1 (H2B-RFP). (**B**) Stills from live imaging of Scramble (top) or *Gpsm2^{1617}* H2B-RFP+ cells (bottom) undergoing planar correction, annotated as in **Figure 1E**. (**C,D**) Cumulative frequency distributions of division orientation from (**C**) Scramble or (**D**) *Gpsm2^{1617}* H2B-RFP (+/-) live imaging experiments at anaphase onset (φ) and one hour later (θ). Scramble RFP+ and *Gpsm2^{1617}* RFP- cells display similar patterns of telophase correction as observed in wild-type explants (**Figure 1G**). While *Gpsm2^{1617}* RFP+ cells are more biased toward planar/oblique at anaphase onset, significant planar correction still occurs; n indicates observed divisions from five embryos imaged in four technical replicates. (**E**) Data from (**C,D**) depicting orientation at anaphase onset (φ) and 1 hr later (θ) for Scramble RFP+ and *Gpsm2^{1617}* RFP- and RFP+ cells. ~ 95% of LGN knockdown cells correct to planar (<30°) 1 hr later. Scale bars, 10 μm. *p<0.05 by Kolmogorov-Smirnov test.

The online version of this article includes the following source data for figure 2:

**Source data 1.** Original measurements used to generate panels C, D, E.

was less predictable (**Figure 1H**). However, we noted that apical daughters undergoing planar telophase correction frequently displayed a unique, balloon-shaped morphology and appeared to maintain contact with the basement membrane (open arrowheads in **Figure 3A**). Remarkably, maintenance of this basal endfoot predicted planar reorientation, while loss of contact predicted the opposite (**Figure 3B,C**; **Figure 3—figure supplement 1A**). Importantly, this correlation between basal contact and telophase correction was unaltered by expression of Scramble or *Gpsm2^{1617}* shRNAs (**Figure 3—figure supplement 1B,C**). These data suggest that transient oblique metaphase-anaphase orientations are corrected in a manner dependent on whether they retain contact with the basement membrane following cleavage furrow ingression.

## Telophase corrective basal contacts display hallmarks of elevated actomyosin contractility

Given the dynamic changes to cell shape that occur during telophase correction, we hypothesized that they may correlate with distinct molecular changes in the underlying actomyosin cytoskeleton. To test this, we performed immunostaining on E16.5 epidermal whole mounts for actin (phalloidin) and active phosphorylated (Ser19) myosin light chain II (pMLC2), and identified rare, oblique divisions with the characteristic basal endfoot. Interestingly, the intensity of pMLC2 was higher specifically in the endfoot process compared to the apical cortex of the same daughter cell, while actin



**Figure 3.** Maintenance of basal contact correlates with planar-directed telophase correction. (**A**) (top) z-projection stills from a movie of a mitotic cell as it enters anaphase (t = 0) through 60 min post-anaphase onset, depicting planar telophase correction. Epidermal-dermal boundary shown by red line. Dividing daughter pairs are outlined with yellow dashed lines. Division orientation angles are shown below (φ, telophase onset; θ, +1h). (bottom), xz *en face* views at same timepoints. Yellow and white arrowheads indicate plane of optical section for apical and basal daughters, respectively. In most cases, planar correction is preceded by maintenance of basement membrane contact (open arrowheads), which are most apparent in the *en face* basal focal plane, where they appear as small membrane circles. (**B**) Data from *Figure 1G,H* sorted based on presence or absence of basal contact. Connecting lines demonstrate that oblique-dividing daughters retaining basal contact correct towards a planar orientation, while those losing contact correct towards perpendicular. (**C**) Data from (**B**) demonstrating that the degree of correction correlates with initial anaphase orientation. (**D**) Whole mount imaging of wild-type E16.5 epidermis stained with phalloidin and phosphorylated myosin-light chain 2 (pMLC2). Orthogonal views (top) of DAPI highlight oblique division orientation. The basal endfoot observed in live imaging of telophase correction (see panel A) can be observed in the basal *en face* view. Pair-wise measurements (inset graph) of pMLC2 at the cell cortex in the apical plane and basal endfoot of oblique divisions are connected by the gray line. (**E**) Cartoon representation of tension-sensitive model of AJ assembly. In the absence of tension, α-E-catenin exists in an autoinhibited closed conformation, masking the α18 epitope. In the presence of actin-mediated tension, α-E-catenin opens, exposing the α18 epitope and vinculin binding domain. (**F**) Whole mount images prepared as in (**D**) stained with total α-E-catenin and open conformation-specific α18 antibody. Pair-wise

*Figure 3 continued on next page*

*Figure 3 continued*

measurements (inset graph) of α18 at the cell cortex in the apical plane and basal endfoot of oblique daughter cells are connected by the gray line demonstrates increased open or 'tensile' α-E-catenin in the basal endfoot. (G) Quantification of α18: α-E-catenin fluorescence intensity ratio in variable division types or stages of mitosis. Anisotropy is greatest in oblique divisions between the basal endfoot and apical cortex of the oblique daughter cell. Scale bars, 10 μm (A,D,F). *P* values determined by Wilcoxon test (D,F). *p<0.05, **p<0.01. See also *Figure 3—figure supplement 1*.
The online version of this article includes the following source data and figure supplement(s) for figure 3:

**Source data 1.** Original measurements used to generate panels B, C, D, F, G.
**Figure supplement 1.** A basal endfoot mediates planar telophase correction.

levels were similar (*Figure 3D*; *Figure 3—figure supplement 1D*). This anisotropy suggests that the basal endfoot may be enriched in contractile actomyosin, which we speculate may serve the function of pulling the apical daughter back into the basal layer.

Increased actomyosin contractility can be indicative of elevated tension across adherens junctions (AJs) that anchor the cytoskeleton to the cell membrane. The AJ is canonically composed of trans-membrane cadherins, which couple neighboring cells through trans-dimerization in the extracellular space and link to the underlying actin-cytoskeleton via α-E-catenin (*Ratheesh and Yap, 2012*). In the presence of actin-dependent tension, α-E-catenin undergoes a conformational change, exposing an epitope within its mechanosensitive modulatory (M) domain that is recognized by the α18 antibody (*Buckley et al., 2014*; *Hansen et al., 2013*; *Rübsam et al., 2017*; *Yonemura et al., 2010*) (*Figure 3E*). To investigate whether α-E-catenin undergoes conformational changes during telophase correction, we performed whole mount immunofluorescence for total and tensile α-E-catenin, seeking out rare anaphase cells undergoing oblique divisions. In agreement with the observed increase in pMLC2 in the basal endfoot of oblique telophase cells, levels of α18 were also higher in the basal endfoot compared to the apical cell cortex in these cells (*Figure 3F*). Importantly, this increased intensity was specific to the α18 epitope as levels of total α-E-catenin did not display similar anisotropy (*Figure 3—figure supplement 1E*). This elevated α18:α-E-catenin ratio was only observed in the basal endfoot of oblique divisions, and not in planar divisions, metaphase cells, or non-mitotic neighbors (*Figure 3G*). These data suggest that increased actomyosin contractility and associated conformational changes to α-E-catenin could play a role in planar directed telophase correction.

## The actin-binding protein, vinculin, regulates α-E-catenin conformation

α-E-catenin (Ctnna1) serves as the core mechanosensor at AJs, such that force across AJs induces a conformational change in α-E-catenin which exposes a vinculin-binding domain within the M region (*Ladoux et al., 2015*; *Yonemura et al., 2010*). The binding of α-E-catenin to both actin and vinculin (Vcl)—another cytoplasmic actin-binding protein that functions at both AJs and focal adhesions—is force dependent, and vinculin and α-E-catenin cooperate to strengthen AJ-mediated adhesion (*Choi et al., 2012*; *Huang et al., 2017*; *Seddiki et al., 2018*; *Thomas et al., 2013*; *Weiss et al., 1998*; *Yao et al., 2014*). Other studies have shown that the actin scaffold afadin (Afdn) is capable of binding directly to α-E-catenin via an internal domain proximal to the vinculin binding domain, and that afadin is recruited to sites of α-E-catenin activation together with vinculin (*Mandai et al., 1997*; *Matsuzawa et al., 2018*; *Pokutta et al., 2002*).

Due to the challenges of finding rare oblique-correcting cells in vivo, to further investigate the interplay between AJ complex proteins and actomyosin contractility, we turned to a calcium-shift adhesion assay in primary cultured keratinocytes (*Vasioukhin et al., 2000*). Following 8h of exposure to 1.5 mM (high) Ca²⁺, Scramble control keratinocytes form linear AJs containing both vinculin and α-E-catenin (*Figure 4A*). *Ctnna1* knockdown led to a reduction in junctional vinculin, while *Vcl* knockdown led to a reduced fluorescence intensity ratio of α18 ("tensile") to total α-E-catenin (*Figure 4A–C*), confirming that the tension sensitivity of α-E-catenin is vinculin-dependent in keratinocytes. Interestingly, while *Vcl* loss reduced the proportion of tensile α-E-catenin, this was a result of a net increase in total α-E-catenin, while total α18 intensity remained unchanged or even increased (*Figure 4—figure supplement 1A*). This suggests that higher levels of junctional α-E-catenin may partially compensate for *Vcl* loss to maintain a threshold level of tensile α-E-catenin.

While α-E-catenin is still recruited to AJs in *Vcl* knockdown keratinocytes, *Vcl*-deficient junctions appeared abnormal, in agreement with a recent report (*Rübsam et al., 2017*). *Vcl*-deficient junctions were wider and more punctate than controls, with a morphology reminiscent of immature "spot AJs" or "adhesion zippers" (*Vasioukhin et al., 2000*). In wild-type keratinocytes cultured for 30 min in high $Ca^{2+}$, nascent cell-cell junctions displayed discontinuous E-cadherin puncta associated with loosely-organized radial actin filaments, while after 8h, E-cadherin and actin became tightly associated in a circumferential belt (*Figure 4—figure supplement 1B*). We developed a quantitative method to measure E-cadherin puncta as a means of assessing junctional maturation, such that higher "continuity" values represent mature linear junctions (e.g., 8h $Ca^{2+}$ shift) while lower continuity values represent spot junctions (e.g., 30 min $Ca^{2+}$ shift) (*Figure 4—figure supplement 1C*).

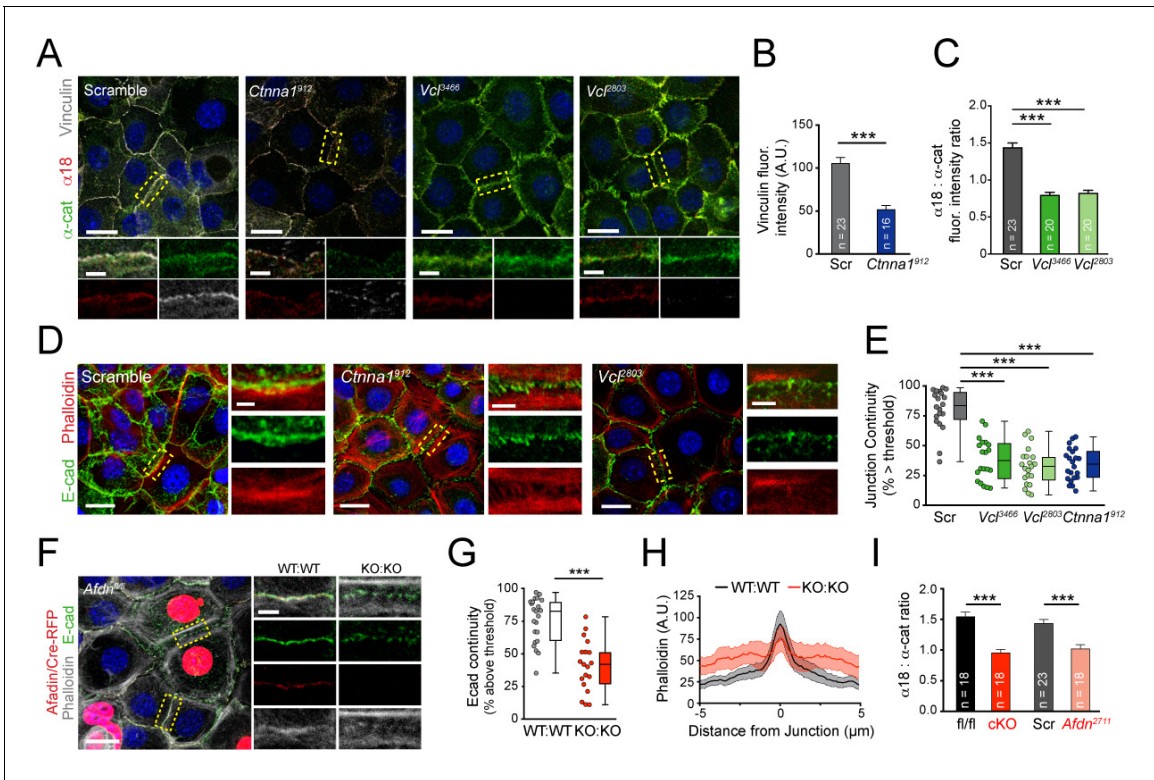

**Figure 4.** Vinculin and afadin regulate α-E-catenin conformation and AJ linkage to the actin cytoskeleton. (A) Stable primary murine keratinocytes cell lines grown in the presence of high (1.5 mM) $Ca^{2+}$ for 8h form nascent cell-cell adhesions, stained for total α-E-catenin (green); open, "tensile" α-E-catenin (α18, red); and vinculin (gray). Single junction magnifications (yellow dashed region) shown below, demonstrate that *Vcl* knockdown results in a reduced α18: α-E-catenin ratio, quantified in (B,C). (B) Fluorescence intensity quantification of junctional vinculin in Scramble and *Ctnna1* knockdown keratinocytes. Loss of *Ctnna1* reduces vinculin accumulation in nascent AJs. (C) Quantification of α18:α-E-catenin fluorescence intensity ratio in Scramble and two independent *Vcl* shRNA cell lines. *Vcl* knockdown reduces the proportion of α-E-catenin in the open conformation. (D) Primary mouse keratinocytes after 8h $Ca^{2+}$ shift—labeled with phalloidin (red) and E-cad (green)—which accumulate in linear bands at cell-cell junctions in Scramble control cells. Yellow boxed region shown at high magnification below; n indicates junctions evaluated. *Vcl* and *Ctnna1* knockdown cells show defects in linear actin accumulation and immature "zipper" junctional morphology. (E) Junction continuity quantification based on % of junction length above threshold for E-cad (see Materials and methods). Loss of *Vcl* or *Ctnna1* reduces junction continuity. (F) *Afdn*^fl/fl primary keratinocytes mosaically infected with Cre-RFP (red) after 8h 1.5 mM $Ca^{2+}$ shift, stained for E-cad (green), afadin (red), and phalloidin (gray). Junctions between two uninfected cells (WT:WT) show linear morphology with consistent E-cad (green), afadin (red) and phalloidin (gray) labeling. In contrast, junctions between two infected cells are punctate, with less junction-associated phalloidin. (G) Quantification of E-cad continuity along junction length, as in (E). (H) Quantification of fluorescence intensity of actin (phalloidin) measured by orthogonal linescans. Phalloidin is decentralized in KO:KO junctions (red) compared to WT:WT (black;. n indicates junctions evaluated. (I) Quantification of α18:α-E-catenin fluorescence intensity ratios from homogenous *Afdn*^fl/fl, *Afdn*-cKO, Scramble, and *Afdn*^2711 primary keratinocytes stained as in (A); n indicates junctions analyzed. Scale bars, 20 μm or 5 μm (junctional insets). *P* values determined by student's unpaired t-test; ***p<0.001. See also *Figure 4—figure supplement 1*.

The online version of this article includes the following source data and figure supplement(s) for figure 4:

**Source data 1.** Original measurements used to generate panels B, C, E, G, H, I.

**Figure supplement 1.** The α-E-catenin/vinculin/afadin complex demonstrates reciprocal regulation to form mature adherens junctions in vitro.

Similar to wild-type controls, following an 8h Ca²⁺ shift, Scramble keratinocytes displayed junctions with linear actin that was closely aligned with E-cadherin. In contrast, in *Ctnna1* and *Vcl* knockdown keratinocytes E-cadherin was punctate and discontinuous, and displaced from the cortical actin belts (**Figure 4D,E**). These data demonstrate that α-E-catenin and vinculin are required for the proper maturation of AJs.

## Afadin is required for normal AJ morphology and is a novel regulator of α-E-catenin conformation

Afadin and its *Drosophila* homolog Canoe (Cno) are required to stabilize actin-AJ associations during moments of high actomyosin contractility, suggesting a role in establishing/maintaining tensile loads (**Choi et al., 2016**; **Sawyer et al., 2011**). To examine whether afadin loss influences AJ-associated actin in keratinocytes, we generated mosaic cultures of wild-type and *Afdn^fl/fl* cells transduced with lentiviral Cre-RFP (**Figure 4F**). E-cadherin+ AJs between two wild-type uninfected cells (WT: WT) showed normal junctional accumulation of afadin, while AJs between two RFP+ cells (KO:KO) lacked afadin (**Figure 4F**, red). *Afdn^fl/fl* Cre-RFP+ cells also demonstrated increased levels of cytoplasmic E-cadherin, and KO:KO junctions displayed punctate, rather than linear, E-cadherin (**Figure 4G**), reminiscent of immature "spot" junctions. In addition, while WT:WT junctions showed tight association of actin with E-cadherin, like *Vcl* and *Ctnna1^912* knockdown AJs, *Afdn* KO:KO AJs showed reduced junctional actin, with actin bundles frequently displaced ~1 μm from the junction (**Figure 4F,H**). These data suggest that afadin plays an essential role in linking cortical actin to the AJ complex, with potential consequences on E-cadherin clustering.

Since it has been shown that AJ components such as E-cadherin regulate junctional recruitment of vinculin from focal adhesions in a tension dependent manner (**Noethel et al., 2018**; **Rübsam et al., 2017**), and we noted that α-E-catenin and vinculin are required for afadin accumulation in the AJ (**Figure 4—figure supplement 1D,E**), we wondered whether afadin reciprocally regulates α-E-catenin or vinculin. Similar to observations in *Vcl* knockdown, knockout or knockdown of *Afdn* resulted in increased junctional accumulation of α-E-catenin, with no observable increase in the α18 epitope, reducing the α18:α-E-catenin fluorescence intensity ratios (**Figure 4I**; **Figure 4—figure supplement 1F–H**). Importantly, loss of *Afdn* also reduced vinculin accumulation in the junction, highlighting a reciprocal regulatory relationship (**Figure 4—figure supplement 1I,J**). Collectively, these data suggest that afadin is a novel regulator of AJ maturation by affecting α-E-catenin conformation and vinculin recruitment.

## *Ctnna1*, *Vcl* and *Afdn* knockdown leads to randomized division orientation

The enrichment of pMLC2 and tensile α-E-catenin in the basal endfoot that we observed in vivo—in addition to the aberrant adhesion and actin organization that we observed in vitro in *Ctnna1*, *Vcl* and *Afdn* mutants—prompted us to investigate whether loss of AJ components alters spindle orientation. To this end, we utilized Survivin to label late-stage mitotic cells and integrin-β4 to label the basement membrane to assess division orientation in E16.5 fixed back skin sections where AJ components where knocked down using our in utero lentiviral delivery method (*Ctnna1*, *Vcl* and *Afdn*), or conditionally knocked out in the epidermis (*Afdn*) (**Figure 5—figure supplement 1A**). We first confirmed the efficacy of knockdown/knockout in vivo using antibodies specific to α-E-catenin, vinculin, and afadin (**Figure 5A-C**). Each AJ protein was localized to the lateral and apical cortex in WT basal cells, as well as to cell membranes in differentiated suprabasal cells. This staining was strongly reduced in RFP+ regions transduced with each shRNA and eliminated in regions where *Afdn* was knocked out by either lentiviral-mediated delivery of Cre-RFP or by conditional deletion using *Krt14^Cre*; *Afdn^fl/fl* (hereafter referred to as *Afdn* cKO) (**Figure 5A-C**).

In each AJ knockdown cohort, we observed a normal bimodal distribution of division angles in wild-type littermate and non-transduced RFP- controls in late stage mitotic cells. However, RFP+ cells displayed randomized division orientation (**Figure 5D–G**), similar to what we observed at anaphase onset in fixed tissue and live imaging. We further validated this phenotype using *Afdn^fl/fl* embryos (**Beaudoin et al., 2012**), and confirmed that division orientation was randomized whether *Afdn* was deleted by lentiviral delivery of Cre-RFP or transgenic expression of *Krt14^Cre*, and analyzed in either sections or wholemounts (**Figure 5H**; **Figure 5—figure supplement 1B,C**). Finally, because

both afadin and vinculin interact directly with α-E-catenin, we sought to test genetically whether afadin and vinculin operate in the same molecular pathway. To do so, we performed embryonic lentiviral injection of the $Vcl^{3466}$ shRNA on an $Afdn$ cKO or $Afdn^{fl/fl}$ background. Examination of division orientation in single and double mutants revealed that vinculin loss did not exacerbate the $Afdn$ cKO phenotype, suggesting that these proteins do not act additively in the context of division orientation (**Figure 5I**).

## Tension-sensitive components of the AJ are essential for telophase correction

We next sought to address whether the randomized division orientation phenotype observed in AJ mutants was due to errors in initial spindle positioning or telophase correction. To this end, we performed live imaging of lentiviral-transduced $Ctnna1^{912}$, $Vcl^{3466}$ and $Afdn^{2711}$ H2B-mRFP1 epidermal explants on a $Krt14^{Cre}$; $Rosa26^{mT/mG}$ background. We began with α-E-catenin, because it had previously been shown that $Ctnna1$ loss leads to randomized division orientation in the developing epidermis (**Lechler and Fuchs, 2005**). As observed earlier with wild-type and $Gpsm2^{1617}$ cells, many $Ctnna1^{912}$ RFP+ basal cells entered anaphase at oblique orientations, with the apical daughter possessing a basal endfoot extending to the basement membrane (**Figure 6A**; **Figure 6—figure supplement 1A**; **Figure 6—video 1**). Because of the high efficiency of transduction achieved with the $Ctnna1$ lentivirus in these experiments, we utilized wild-type littermates as controls rather than RFP-cells, which were rare. We imaged 74 $Ctnna1$ RFP+ mitotic cells, and observed that α-E-catenin loss had no effect on initial anaphase orientation, which was randomized, akin to wild-type littermates. However, while wild-type cells corrected to a bimodal distribution within 1h of anaphase onset, there was no change in the distribution of division angles in $Ctnna1$ RFP+ cells between anaphase onset and 1h later (**Figure 6B**). Whether or not apical daughters maintained basal contact, there appeared to be no obvious pattern to the directionality of telophase reorientation, with a majority of cells showing little or no change over 1h following anaphase onset (**Figure 6C**; **Figure 6—figure supplement 1B,F**).

Like $Ctnna1^{912}$ knockdown cells, $Vcl^{3466}$ RFP+ cells frequently entered anaphase at oblique orientations and showed little movement during telophase (**Figure 6—figure supplement 1C**; **Figure 6—video 2**). RFP- cells corrected to a bimodal distribution, although in these experiments a higher proportion of planar corrections were observed than in previous studies, perhaps due to the slight differences in their developmental stage (**Figure 6D**). Nevertheless, as a population, $Vcl^{3466}$ RFP+ cells displayed a randomized distribution of division angles at both anaphase onset and 1h later (**Figure 6D**). As with $Ctnna1$ loss, $Vcl$ knockdown reduced the magnitude of telophase reorientation, and eliminated the predictiveness of basal contacts for correction directionality, causing failure in both perpendicular and planar correction (**Figure 6E**; **Figure 6—figure supplement 1D,F**).

In *Drosophila*, the afadin homologue Cno is essential for asymmetric cell division of embryonic neuroblasts (**Speicher et al., 2008**). Recent studies in mammals have similarly described a role for afadin in regulating division orientation in the embryonic kidney and cerebral cortex (**Gao et al., 2017**; **Rakotomamonjy et al., 2017**). In fixed tissue, we knocked down or knocked out $Afdn$ by three different methods, each resulting in randomized division orientation in Survivin+ late-stage mitotic cells (**Figure 5G,H**; **Figure 5—figure supplement 1**). Because the native fluorescence of Cre-RFP is dim and photobleaches rapidly, and live imaging of $Afdn$ knockouts requires a complex breeding scheme involving four alleles, we utilized the $Afdn^{2711}$ shRNA for ex vivo imaging experiments (**Figure 6F**). As with α-E-catenin and vinculin loss, $Afdn$ knockdown had no effect on initial anaphase orientation, while oblique divisions failed to undergo either planar or perpendicular-directed telophase correction (**Figure 6G,H**; **Figure 6—figure supplement 1E,F**; **Figure 6—video 3**). $Afdn$ knockdown phenocopied loss of vinculin and α-E-catenin, with minimal or randomized reorientation of oblique divisions (**Figure 6I**). Moreover, while endfoot contact at anaphase onset was predictive of telophase correction directionality in RFP- cells, this was not the case in $Afdn^{2711}$ RFP+ cells (**Figure 6J**). Notably, however, while oblique $Ctnna1^{912}$ and $Vcl^{3466}$ cells generally retained basal endfoot processes if they were present at anaphase onset, 73% of oblique $Afdn^{2711}$ cells lost contact during telophase, suggesting that afadin may function in endfoot retention. Collectively, these studies demonstrate that mechanosensitive AJ proteins do not appear to function in initial spindle positioning, but play important modulatory roles in mediating telophase correction, which, when disrupted, lead to persistent division orientation errors.

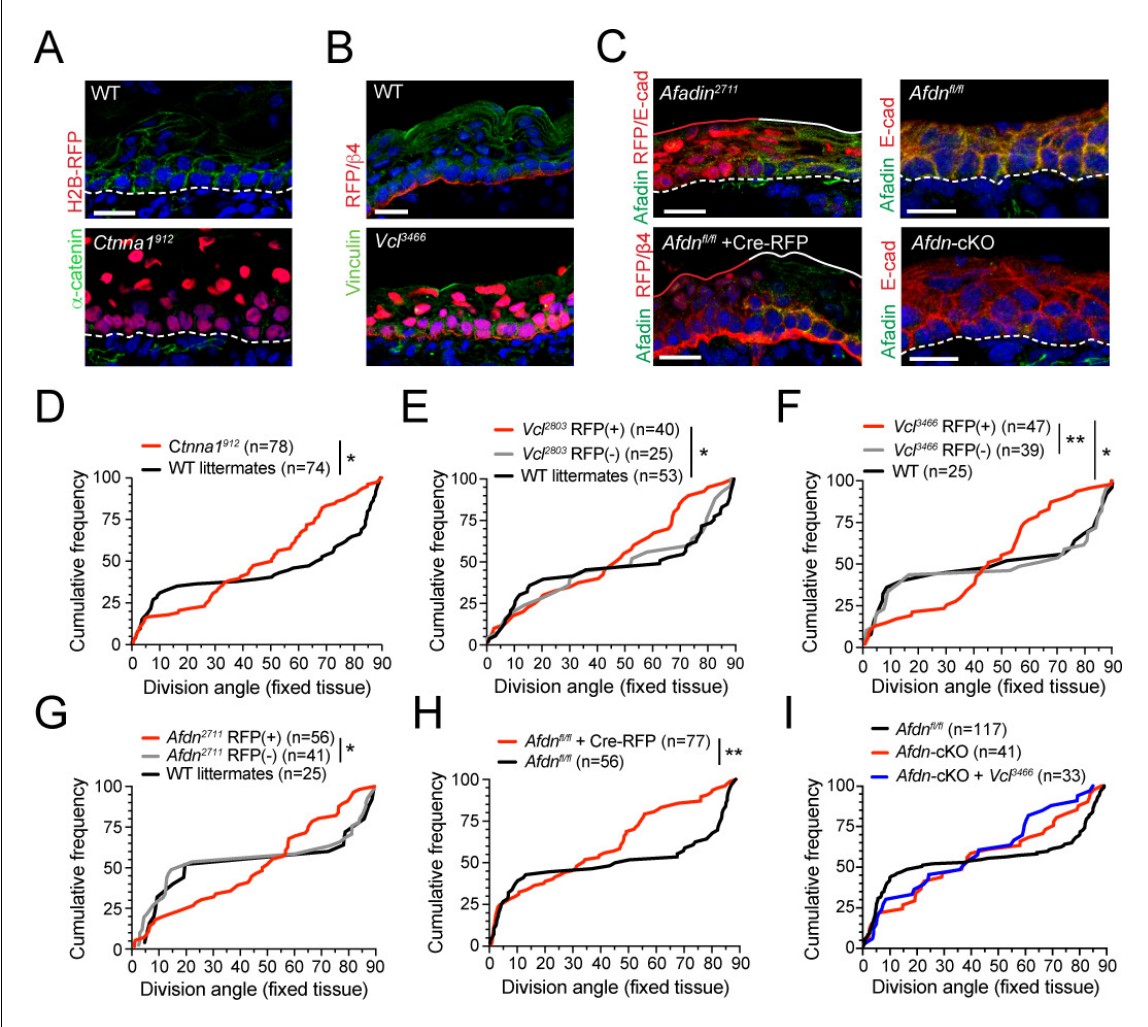

**Figure 5.** The α-E-catenin/vinculin/afadin pathway is required for normal division orientation. (**A**) Immunofluorescent images taken from E16.5 sagittal sections of wild-type littermate controls (left) or embryos transduced with *Ctnna1^912* H2B-RFP (right). Epidermal junctional α-E-catenin (green) is lost in *Ctnna1^912* RFP+ epidermis. (**B**) E16.5 epidermis infected with *Vcl^3466* H2B-RFP (red) and stained rabbit with anti-vinculin antibody. While suprabasal staining is dramatically reduced in infected samples, some non-specific cytoplasmic basal-layer staining remains. (**C**) Afadin (green) and E-cadherin (red) immunostaining in E16.5 sections. Mosaic region of *Afdn^2711* H2B-RFP (top panel) or Cre-RFP (in *Afdn^fl/fl* embryo; bottom panel) lentiviral transduction. Region of high transduction (red line) demonstrates efficient loss of junctional afadin signal, spared in region of low transduction (white line). E16.5 *Afdn^fl/fl* controls (right, top) with conditional deletion mediated by *Krt14^Cre* (cKO) (right, bottom). (**D–I**) Cumulative frequency distributions of telophase division angles from fixed E16.5 sections of shRNA knockdown samples and littermate controls. (**D**) *Ctnna1^912* knockdown (red) and control littermates (black); *n* indicates measurements from 6 to 7 independent embryos. (**E**) *Vcl^2803* H2B-RFP mosaic samples showing RFP+ mutants (red) alongside RFP-internal (gray) and wild-type littermate (black) controls; *n* indicates measurements from 3 to 4 independent embryos. (**F**) *Vcl^3466* H2B-RFP mosaic samples shown as in E; *n* indicates measurements from 3 to 4 independent embryos. (**G**) *Afdn^2711* H2B-RFP mosaic samples shown as in E-F; *n* indicates measurements from 3 to 6 independent embryos. (**H**) *Afdn^fl/fl* Cre-RFP samples (red) shown alongside uninjected littermates (black); *n* indicates measurements from 3 to 4 independent embryos. (**I**) Cumulative frequency distribution of E16.5 telophase division angles in *Afdn^fl/fl*, *Afdn* cKO, and *Afdn* cKO + *Vcl^3466* H2B-RFP epidermis. Vinculin knockdown does not exacerbate *Afdn* knockout phenotype. Scale bars, 20 μm (**A–C**). *P* values determined by Kolmogorov-Smirnov test (**D–I**). *p<0.05, **p<0.01. See also *Figure 5—figure supplement 1*.

The online version of this article includes the following source data and figure supplement(s) for figure 5:

**Source data 1.** Original measurements used to generate panels D, E, F, G, H, I.

**Figure supplement 1.** AJ loss-of-function mutants display errors in division orientation.

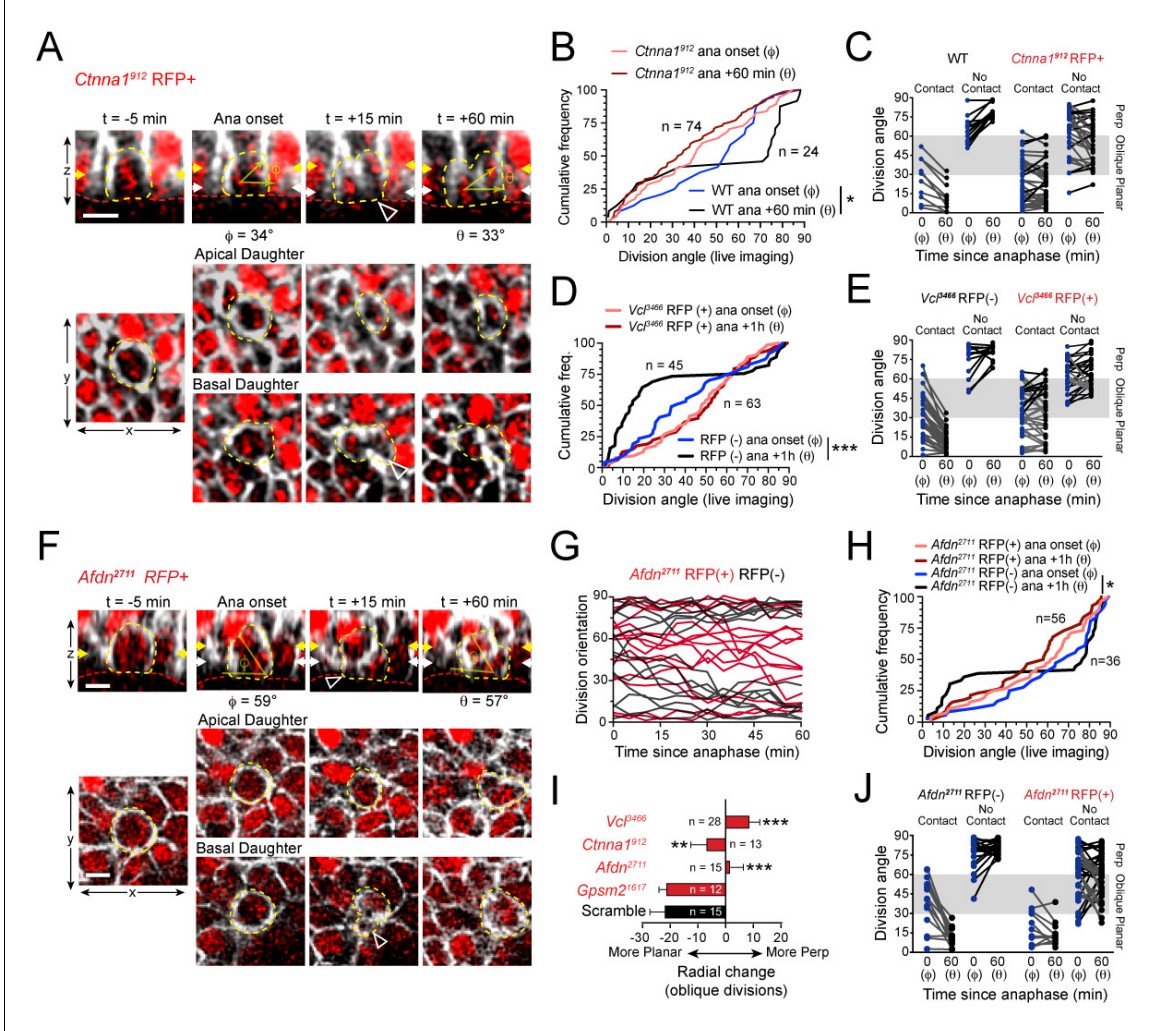

**Figure 6.** AJ mutants fail at both planar and perpendicular telophase correction. (**A**) Movie stills of *Ctnna1^912* RFP+ mitotic cell, annotated as in *Figure 3A*. While the presence of basal contact (open arrowhead) would predict planar correction, this division remains oblique when reevaluated 1h later. (**B**) Cumulative frequency distribution of division angles from E16.5 live imaging experiments of *Ctnna1^912* RFP+ and wild-type littermates; *n* values indicate cells imaged from three embryos images in two separate sessions. (**C**) Division orientation at anaphase onset (φ) and 1h later (θ) for *Ctnna1^912* knockdown and wild-type littermates, plotted from data in (**B**). *Ctnna1^912* RFP+ cells show no obvious correction pattern. (**D**) Cumulative frequency distribution of division orientation at anaphase onset (φ) and 1hrpost-anaphase (θ) for RFP+ and RFP- populations, from movies of *Vcl^3466* mosaic tissue; *n* indicates divisions from four embryos imaged in three separate sessions. (**E**) Data from (**D**) depicting orientation at anaphase onset (φ) and 1h later (θ) for RFP- and RFP+ cells. RFP- controls sort anaphase orientation (φ) into bimodal distribution within 1h(θ) in a basal-contact dependent manner; *Vcl^3466* RFP+ cells display minimal change, or correct irrespective of basal contact. (**F**) An obliquely-oriented *Afdn^2711* RFP+ cell fails to reorient, while losing basal contact (open arrowhead). (**G**) Timelines of division orientation at 5 min intervals from movies of *Afdn^2711* RFP- (black) and RFP+ (red) for 15 representative cells per group. Telophase reorientation establishes bimodal distribution within ~30 min in RFP- control cells that enter anaphase at oblique angles, while RFP+ cells fail to demonstrate any sorting behavior over a full hour following anaphase onset. (**H**) Cumulative frequency distributions of division orientation from E16.5 live imaging of *Afdn^2711* RFP+ and wild-type littermates; *n* indicates observed divisions from three embryos imaged in two separate sessions. (**I**) Radial change (φ-θ) for oblique anaphase divisions (30°−60°) in several shRNA conditions. While loss of LGN allows for normal telophase correction, *Afdn*, *Ctnna1*, and *Vcl* knockdown results in incoherent or minimal radial change; *n* indicates number of divisions from 3 to 6 individuals embryos images in 2–4 technical replicates. (**J**) Division orientation at anaphase onset (φ) and one hour later (θ) for *Afdn^2711* RFP+ and RFP- cells, plotted from data in (**C**). RFP- controls correct into a bimodal distribution, while RFP+ cells reorient randomly. Scale bars, 10 μm. *P* values determined by Kolmogorov-Smirnov test (**B,D,H**) or student's t-test (**I**). *p<0.05, **p<0.01, ***p<0.001. See also *Figure 6—figure supplement 1*.

The online version of this article includes the following video, source data, and figure supplement(s) for figure 6:

**Source data 1.** Original measurements used to generate panels B, C, D, E, G, H, I, J.

**Figure supplement 1.** AJ loss-of-function mutants display errors in division orientation.

**Figure 6—video 1.** Persistent oblique division in *Ctnna1^912* knockdown mitotic basal cell.

*Figure 6 continued on next page*

*Figure 6 continued*

https://elifesciences.org/articles/49249#fig6video1

**Figure 6—video 2.** Persistent oblique division in *Vcl*^*3466* knockdown mitotic basal cell.

https://elifesciences.org/articles/49249#fig6video2

**Figure 6—video 3.** Persistent oblique division in *Afdn*^*2711* knockdown mitotic basal cell.

https://elifesciences.org/articles/49249#fig6video3

## Telophase correction occurs independently of canonical polarity and spindle-orienting cues

The *Drosophila* afadin ortholog Cno is essential for early establishment of apical-basal polarity during cellularization (*Bonello et al., 2018*; *Choi et al., 2013*). A similar role has been described for afadin in mammalian development (*Komura et al., 2008*; *Rakotomamonjy et al., 2017*; *Yang et al., 2013*). Furthermore, both Par3 and its *Drosophila* ortholog Bazooka are required for oriented cell divisions via regulation of LGN localization (*Schober et al., 1999*; *Williams et al., 2014*; *Wodarz et al., 1999*). Thus, we asked whether *Afdn* loss impacts expression of the canonical apical polarity cue Par3. In *Afdn*^*fl/fl* controls, Par3 accumulates at the apical cortex throughout the cell cycle (*Figure 7—figure supplement 1A*). We measured Par3 radial fluorescence intensity at interphase and determined that *Afdn* cKO epidermis shows a 15–30% reduction in apical accumulation (*Figure 7—figure supplement 1A,B*). However, this had no effect on the apical positioning of centrosomes (*Figure 7—figure supplement 1C,D*), suggesting that apical-basal polarity remains largely intact in *Afdn* mutants.

Previous studies have shown that Cno interacts directly with the LGN ortholog Pins and regulates its cortical recruitment (*Speicher et al., 2008*; *Wee et al., 2011*). Mammalian afadin and LGN also directly interact in HeLa cells, where they function to promote planar divisions (*Carminati et al., 2016*). In addition, E-cadherin is capable of regulating division orientation through a direct interaction with LGN (*Gloerich et al., 2017*; *Hart et al., 2017*). Finally, *Ctnna1* knockout has been reported to perturb LGN localization in epidermal basal cells (*Lechler and Fuchs, 2005*). These studies suggested that the division orientation defects we observed in *Ctnna1* and *Afdn* mutants could be due to mislocalized LGN.

Using pHH3 to label cells in early mitosis in *Afdn*^*fl/fl* controls and *Afdn* cKO mutants, we observed similar patterns of LGN crescent localization, cortical intensity, and efficiency of apical polarization (*Figure 7A–D*). In addition, we did not observe any obvious or significant changes to LGN localization in *Afdn*^*2711*, *Ctnna1*^*912* or *Vcl*^*3466* cells (*Figure 7C,D*; *Figure 7—figure supplement 1E*). Thus, AJ components appear to be dispensable for initial apical positioning of LGN. In *Drosophila* neuroblasts, genetic epistasis and protein localization studies support the view that Cno/afadin acts downstream of Pins/LGN and upstream of Mud/NuMA (*Speicher et al., 2008*). However, we find that neither NuMA, nor its downstream binding partner dynactin, appears to be mislocalized in *Afdn* mutants (*Figure 7—figure supplement 1F–H*). In addition, NuMA staining overlapped with LGN in early mitotic cells, regardless of afadin presence/absence (91% in *Afdn*^*fl/fl*, n = 22; 93% in *Afdn* cKO, n = 14). These data suggest that afadin plays little, if any role, in regulating the LGN-NuMA-dynactin pathway during initial spindle positioning.

We previously demonstrated that the mitotic spindle can become misaligned with cortical LGN during metaphase, for example following NuMA (*Numa1*) knockdown (*Williams et al., 2011*). Thus, we sought to examine whether *Afdn* loss could also lead to uncoupling of the division axis from LGN polarity cues during mitosis, perhaps independently of NuMA. To test this, we co-stained prefixed E16.5 *Afdn*^*fl/fl* and *Afdn* cKO sections with LGN and α-tubulin in order to visualize spindles during metaphase, and cleavage furrow ingression later in telophase (*Figure 7E*). Importantly, while afadin loss altered telophase division orientation, it had no effect during metaphase, where spindles were randomly-oriented (*Figure 7—figure supplement 2A*). Furthermore, the apical LGN crescent aligned with the metaphase spindle axis—regardless of its orientation—in both *Afdn* cKO and control embryos (*Figure 7E,F*; *Figure 7—figure supplement 2B*). By all these metrics, LGN localization was also unperturbed in *Afdn*^*2711*, *Ctnna1*^*912*, *Vcl*^*2803* and *Vcl*^*3466* knockdowns as well (*Figure 7C–F*). However, in telophase cells, while LGN remained apically-positioned in both controls and *Afdn* mutants, the orientation of the spindle axis became uncoupled from LGN in *Afdn* mutants

(*Figure 7F*; *Figure 7—figure supplement 2B,D*). Similarly, knockdown of α-E-catenin or vinculin phenocopied afadin loss, demonstrating that AJ perturbation does not alter LGN localization, but does affect the ability of telophase cells to reorient in response to apical cues (*Figure 7F*; *Figure 7— figure supplement 2C,D*).

These findings, together with our observation that AJ and LGN mutants differ in their planar telophase correction phenotypes, suggest that afadin, α-E-catenin and vinculin act independently of LGN in the context of spindle orientation. As further evidence, we find that while LGN strongly colocalizes with known binding partners Gαi3 and Insc, afadin demonstrates minimal colocalization with LGN either pre- or post-chromosome segregation (*Figure 7—figure supplement 2E*). Finally,

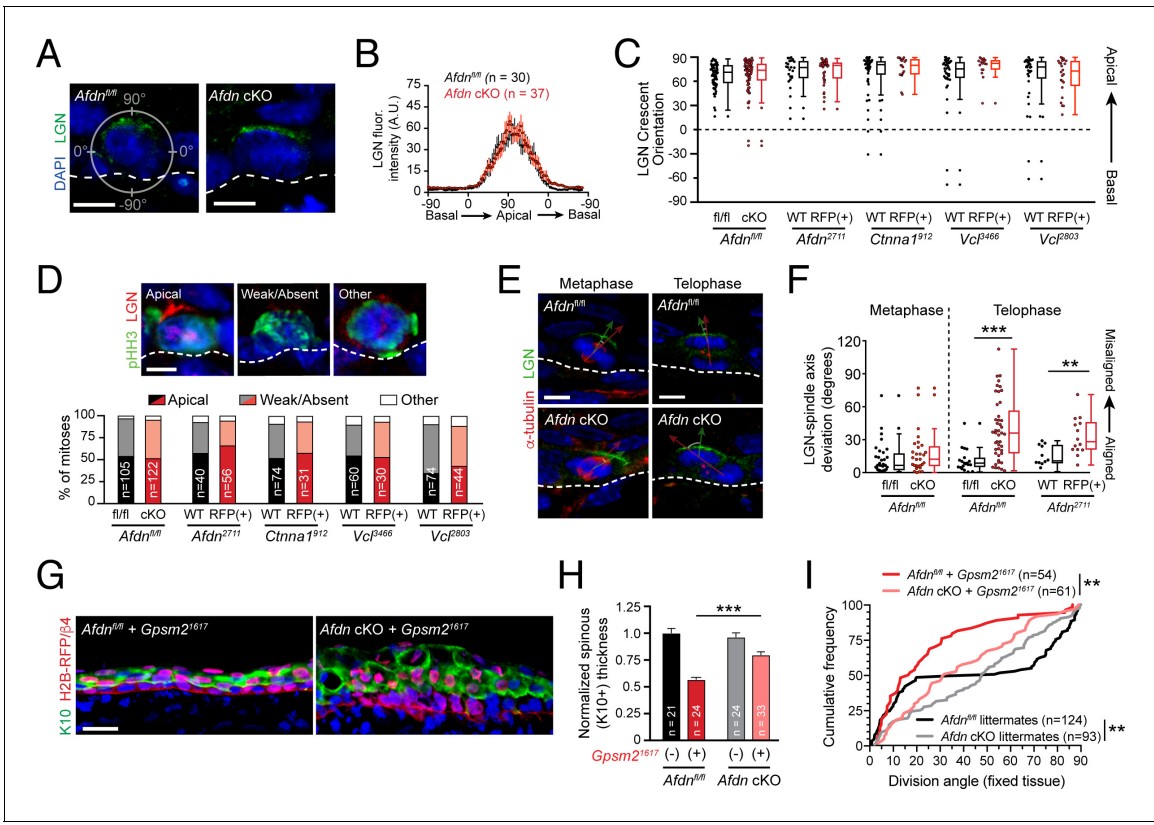

**Figure 7.** AJ mutants alter division orientation via LGN-independent mechanisms. (**A**) Immunostaining for LGN (green) in E16.5 *Afdn^fl/fl* and *Afdn*-cKO epidermis. LGN localizes at the apical cortex during mitosis regardless of afadin loss. (**B**) Quantification of LGN radial fluorescence intensity in E16.5 mitotic cells; *n* indicates LGN+ mitoses from 2 to 3 independent embryos. (**C**) Orientation of LGN crescents in E16.5 mitotic cells from indicated groups. Knockdown or knockout of AJ components does not significantly alter the tendency of LGN to localize apically. (**D**) (top) LGN (red) localization patterns in mitotic (green) basal keratinocytes. (Bottom) Quantification of LGN rate of recruitment, binned by genotype. LGN localizes to the apical cortex in ~50% of mitoses (black/red), is absent in ~45% (gray/pink), and "other" in the remaining ~5% (white), remaining unchanged in AJ knockdown/ knockout mutants; *n* indicates mitotic cells from 2 to 3 independent embryos. (**E**) Costaining of E16.5 metaphase (left) and telophase (right) divisions with α-tubulin (red) and LGN (green) in *Afdn*-cKO (bottom) and *Afdn^fl/fl* control littermates (top). (**F**) Quantification of the deviation between the metaphase spindle or division axis (red arrow in E) and LGN radial orientation (green arrow in E). *Afdn* knockout does not disrupt early spindle-LGN linkage, but shows oblique telophase orientation despite normal localization of LGN. (**G**) Immunostaining for the differentiation marker K10 (green) and lentiviral H2B-RFP reporter simultaneously with β4-integrin (red) in E16.5 *Gpsm2^1617* infected embryos with an *Afdn^fl/fl* (left) or *Afdn*-cKO (right) background. Dual loss of *Afdn* and *Gpsm2* results in increased stratification relative to *Gpsm2* loss alone. (**H**) Quantification of spinous layer (K10+) thickness from images as in (G). (**I**) Cumulative frequency distribution of telophase division angles from fixed sagittal sections of E16.5 embryos. *n* indicates number of divisions from 2 to 3 independent embryos. Scale bars, 5 μm (A,D,E), 25 μm (G). **p<0.01, ***p<0.001, determined by Kolmogorov-Smirnov test (I) or student's t-test (F,H). See also *Figure 7—figure supplements 1* and *2*.

The online version of this article includes the following source data and figure supplement(s) for figure 7:

**Source data 1.** Original measurements used to generate panels B, C, D, F, H, I.

**Figure supplement 1.** *Afdn* loss-of-function does not affect functional apicobasal polarity or downstream components of spindle orientation.

**Figure supplement 2.** AJ components alter division orientation in an LGN-independent manner.

there are several contexts during epidermal development where LGN is not required for division orientation. First, although hair placode progenitors undergo perpendicular asymmetric divisions (*Ouspenskaia et al., 2016*), LGN is weakly expressed and is not required for proper division orientation in mitotic placode cells (*Byrd et al., 2016*). Second, while LGN loss reduces perpendicular divisions in the interfollicular epidermis at E16.5, LGN is dispensable at E14.5, when the majority of divisions are planar and LGN is rarely cortical (*Williams et al., 2014*). Conversely, *Afdn* knockdown increases the frequency of oblique divisions in both contexts, suggesting an LGN-independent function for afadin in both perpendicular and planar divisions (*Figure 7—figure supplement 2F,G*). Together, these data suggest that afadin is a minor or transient LGN-interactor in vivo and support a polarity- and LGN-independent role for afadin in telophase correction.

## Telophase correction and early mitotic spindle orientation function as parallel pathways

We next sought to test genetically whether the telophase correction pathway can override the initial spindle positioning cues provided by LGN. To address this, we generated afadin and LGN dual loss-of-function embryos by injecting the $Gpsm2^{1617}$ lentivirus into either a wild-type ($Afdn^{fl/fl}$) or *Afdn* cKO background. While loss of LGN alone recapitulated the previously described phenotype of impaired stratification (*Williams et al., 2011*), loss of *Afdn* on an *Gpsm2* mutant background partially rescued this differentiation defect (*Figure 7G,H*). Moreover, the predominantly planar division orientation observed in *Gpsm2* single mutants became more randomized upon dual loss with *Afdn*, generating an intermediate phenotype (*Figure 7I*). These epistasis experiments suggest that telophase correction operates in parallel with, rather than downstream of, the canonical spindle orientation pathway.

Of note, double mutants largely lacked perpendicular (70°−90°) divisions, further supporting a specific role for the LGN complex to generate this division type. Taken together, these data suggest that early spindle orientation cues direct imprecise perpendicular divisions in an LGN-dependent manner. These divisions are then refined into the characteristic bimodal pattern of perpendicular or planar divisions by telophase correction. However, these data also suggest that LGN-directed perpendicular correction is still dependent on the AJ components driving telophase correction.

## Telophase correction also occurs during early stratification

The observations that afadin is required for telophase correction at E16.5 (*Figure 6F-J*), and that *Afdn* mutants display division orientation defects at both early and peak stages of stratification (*Figure 5G-I*; *Figure 7—figure supplement 2G*) prompted us to examine whether telophase correction occurs throughout epidermal morphogenesis. Thus, we performed live imaging on wild-type $Krt14^{Cre}$; $Rosa26^{mT/mG}$ epidermal explants at E14.5, when stratification initiates (*Figure 8A*). Even though nearly all divisions at E14.5 are planar (*Lechler and Fuchs, 2005*; *Williams et al., 2014*), remarkably, in wild-type cells at telophase onset, the distribution of observed orientations was randomly-distributed, similar to what was observed at E16.5 (*Figure 8B*; compare to Figure 1G). However, while 47% of cells (n=78) entered telophase oriented obliquely, the vast majority of these possessed a basal endfoot and corrected to planar within 1h of telophase onset (*Figure 8C*). On the other hand, the few cells (28%) that did not maintain basal contact corrected randomly at E14.5, in contrast to E16.5, when they invariably corrected to perpendicular (compare *Figure 8C to 3B*). Since LGN does not localize cortically or influence division orientation at E14.5 (*Williams et al., 2014*), this provides additional evidence that LGN is necessary for perpendicular telophase correction.

## Telophase correction impacts cell fate decisions

At E14.5, *Afdn* mutants displayed fewer planar and more oblique divisions compared to controls (*Figure 7—figure supplement 2G*), which led us to ask whether afadin loss could promote precocious differentiation. In E14.5 $Afdn^{2711}$ mosaic epidermis, we noted that Keratin-10 (K10)—a marker of differentiated cells—was enriched in RFP+ mutant regions compared to RFP- wild-type regions (*Figure 8D*). While basal cell density was similar between $Afdn^{2711}$ embryos and non-transduced littermates, the density of differentiated cells—whether assessed by their suprabasal (SB) position or K10 expression—was significantly higher in *Afdn* mutants (*Figure 8E*, *Figure 8—figure supplement 1A*). This was unlikely to be caused by hyperproliferation because similar levels of mitotic cells were

observed at both E14.5 and E16.5 in *Afdn²⁷¹¹* RFP+ and wild-type littermate controls (*Figure 8—figure supplement 1B*). Like afadin, loss of α-E-catenin resulted in a hyperstratified epidermis and increased suprabasal cell density (*Figure 8F*; *Figure 8—figure supplement 1C*). Consistent with previous observations in E18.5 *Ctnna1* knockout epidermis (*Beronja et al., 2010*; *Vasioukhin et al., 2001*), the precocious differentiation observed in these mutants persisted into later ages (*Figure 8—figure supplement 1D, E*). Notably, in contrast to a previous report that late embryonic *Ctnna1* epidermis is hyperproliferative (*Vasioukhin et al., 2001*), we do not observe any elevation in mitotic cells in either E14.5 or E16.5 *Ctnna1* epidermis (*Figure 8—figure supplement 1F*), which is more in agreement with a recent study that showed a mild increase in BrdU+ cells but net growth disadvantage of *Ctnna1⁹¹²* basal cells (*Beronja et al., 2010*). Thus, we feel it is more likely that the precocious

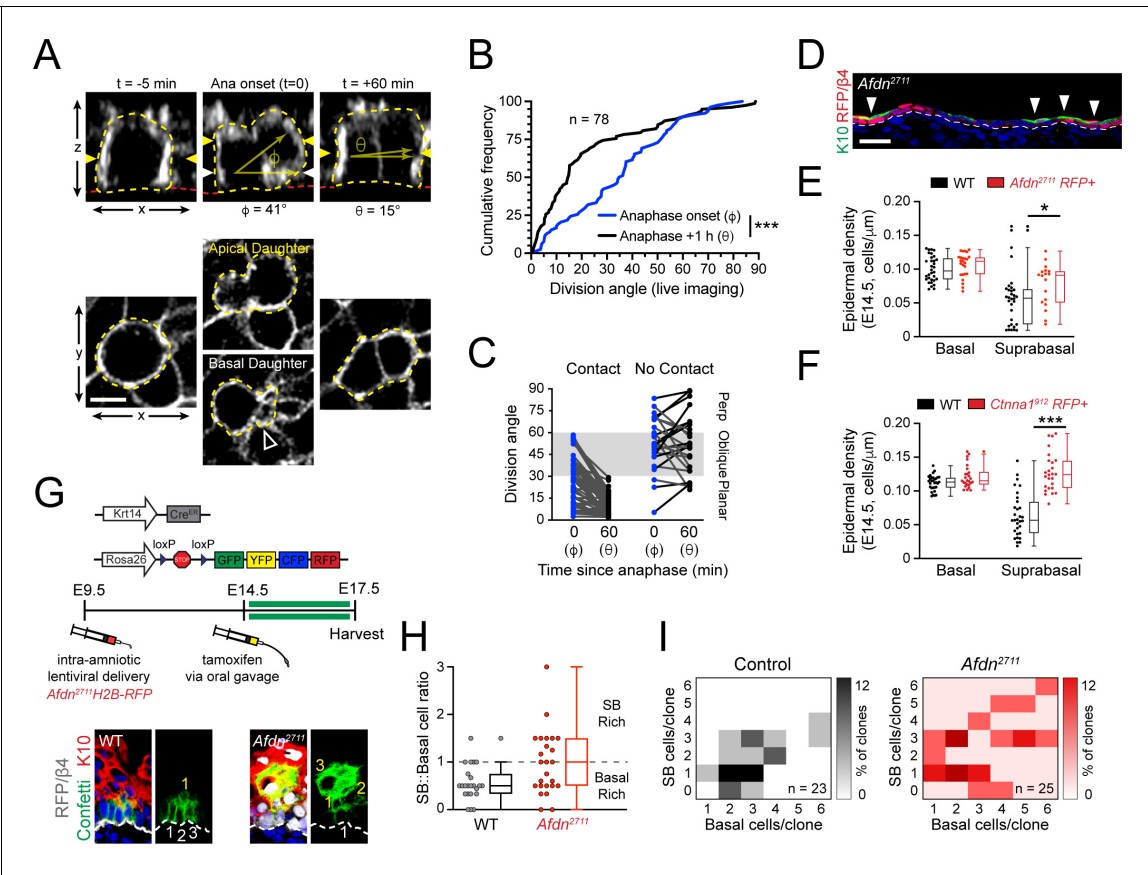

**Figure 8.** Planar telophase correction limits precocious and sustained differentiation and biases clones towards symmetric cell divisions. (**A**) (top) z-projection stills from a movie of an E14.5 mitotic cell, annotated as in *Figure 1E*. (**B**) Cumulative frequency distribution of division angles from live imaging experiments of E14.5 embryos at anaphase onset (blue; φ) and 1h later (black; θ); *n* indicates number of divisions from 3 embryos across two independent sessions. (**C**) Data from (B) depicting division orientations at telophase and 1h later, sorted based on retention/loss of basal contact throughout cell division. Connecting lines demonstrate that, at E14.5, planar correction occurs in a contact dependent manner, while mitoses that lose contact demonstrate no obvious pattern of correction. (**D**) Sagittal section of E14.5 epidermis with mosaic *Afdn²⁷¹¹* H2B-RFP transduction. Regions of high infection display increased stratification, as demonstrated by K10 (green) positivity. (**E-F**) Quantification of epidermal differentiation from E14.5 sagittal sections. *Afdn* (E) or *Ctnna1* (F) knockdown increases suprabasal cell density, suggesting precocious differentiation. (**G**) (top) Graphical depiction of clonal lineage tracing strategy; (bottom) Representative images of E17.5 sagittal sections from lineage tracing experiments stained with GFP (green), K10 (red), and RFP/β4-integrin (gray). *Afdn²⁷¹¹* knockdown clones display asymmetric (suprabasal) bias. (**H**) Clonal density arrays representing all evaluated clones (except delamination events) from experiments outlined in (**G**). The proportion of total clones for each possible combination of basal/suprabasal cells is coded on a color spectrum correlating to 0–12% of all clones. (**I**) Quantification of suprabasal (SB):basal cell ratio for individual clones. Knockdown of *Afdn* results in a higher ratio of SB cells in individual clones compared to wild-type littermates. Scale bars, 5 μm (A), 25 μm (D). *p<0.05, ***p<0.001, determined by Kolmogorov-Smirnov test (B) or student's t-test (F). See also *Figure 8—figure supplement 1*. The online version of this article includes the following source data and figure supplement(s) for figure 8:

**Source data 1.** Original measurements used to generate panels B, C, F, H , I.
**Figure supplement 1.** Failed telophase correction induces precious, sustained hyperstratification without impacting proliferation or delamination.

differentiation observed in *Ctnna1*[912] mutants is due to persistent oblique divisions caused by errors in telophase correction, rather than to hyperproliferation. Collectively, these data suggest that telophase correction influences differentiation throughout epidermal development.

While the previous experiments demonstrated that AJ loss alters both division orientation and promotes differentiation, they do not address whether telophase correction errors directly impact cell fate choices. To explore whether afadin loss alters fate decisions, we performed short term (72h) lineage tracing experiments using *Krt14*[CreER]; *Rosa26*[Confetti] reporter mice in combination with *Afdn*[2711] knockdown, and examined the number of progenitor (basal) and differentiated (SB, K10+) progeny within resultant clones (*Figure 8G*). The lentiviral shRNA strategy was chosen to target *Afdn* because the alternative—*Krt14*[CreER]-mediated deletion of the *Afdn*[fl] allele—would result in *Afdn* deletion occurring simultaneous with, rather than prior to, clonal induction. Moreover, in contrast to lentiviral-delivered *Afdn*[2711], *Krt14*[Cre]-mediated deletion of *Afdn* did not cause obvious differentiation defects (compare *Figure 7H* to *Figure 8—figure supplement 1E*), likely because *Afdn* deletion occurs later with *Krt14*[Cre] (*Beronja et al., 2010*).

We administered a single dose of tamoxifen at E14.5 by oral gavage, then harvested embryos at E17.5, when we analyzed clones obtained from *Afdn*[2711] RFP+ and uninjected (wild-type) littermates. In agreement with our mitotic index measurements, knockdown of *Afdn* did not alter the distribution of clone sizes (*Figure 8—figure supplement 1G*). However, *Afdn*[2711] clones frequently displayed a greater proportion of suprabasal cells per clone when compared to uninjected littermates (*Figure 8*; *Figure 8—figure supplement 1H*). We utilized clonal density arrays to display clone size distributions, such that basal and SB cells/clone are plotted on x and y axes, respectively, and darker colors indicate higher frequencies of specific clone types (*Byrd et al., 2019*). These data demonstrate that *Afdn*[2711] contain a higher proportion of SB-rich clones (*Figure 8I*).

We further characterized clones into four subtypes: (1) balanced (1:1 ratio of basal:SB cells), (2) basal-rich (basal:SB ratio>1), (3) SB-rich (basal:SB ratio<1), and delamination (basal cells=0). Delamination is an alternative differentiation mechanism to asymmetric cell division, whereby a basal cell detaches from the underlying basement membrane and initiates differentiation without dividing. We previously showed through lineage tracing that delamination drives the initial phase of stratification, while asymmetric cell divisions predominate during peak stratification (*Williams et al., 2014*). While we did not observe clear delamination events in our 3-6 hour live imaging experiments, genetic lineage tracing revealed that a similar and significant fraction of clones in both WT and *Afdn*[2711] epidermis (48% vs 40%) arose from delamination (*Figure 8—figure supplement 1I*). A comparison of the mitotic clone distribution between WT and *Afdn*[2711] clones revealed that *Afdn*[2711] epidermis contains a much greater number of SB-rich clones (24% vs 2% in WT), at the expense of the basal-rich (29% vs 41%) subtype (*Figure 8—figure supplement 1I*). Since delamination events slightly decrease in *Afdn* mutants compared to WT controls, this further suggests that the excess differentiation observed in *Afdn* mutants is attributable to an increase in asymmetric cell divisions rather than compensatory delamination.

We conclude that the excess oblique divisions observed in *Afdn* mutants, which fail to be corrected during telophase—impacts cell fate decisions, favoring differentiation over self-renewal. This further implies that a high proportion of oblique divisions are operationally asymmetric. In conclusion, we provide several lines of evidence that telophase correction contributes to establishing proper epidermal architecture: (1) the tensile AJ components afadin, α-E-catenin and vinculin fail to correct during telophase, leading to a persistent excess of oblique divisions, (2) AJ mutants which fail at telophase correction induce excess stratification, and (3) the failure of oblique divisions to correct to planar during telophase leads to a bias toward differentiation over self-renewal.

## Discussion

### A two-step mechanism for division axis determination

These studies shed new light on the mechanisms governing oriented cell divisions in the developing epidermis and identify telophase correction as an important contributor to balancing symmetric and asymmetric divisions throughout stratification. While previous studies have demonstrated essential roles for canonical spindle orientation genes in division orientation, we now show that initial spindle positioning is only one part of the process (*Figure 9*). Our data suggest that LGN and associated

proteins operate early in mitosis to promote perpendicular divisions, but do so with a high degree of imprecision, resulting in a wide distribution of anaphase division angles. While this function of LGN is required for perpendicular divisions to occur, this fails to explain the bimodal distribution of division angles observed in telophase. In the second phase of our model, telophase cells undergo dynamic reorientation towards a planar or perpendicular orientation, where the direction of correction is dependent on contact with the basement membrane via a basal endfoot. We further demonstrate that LGN is also required for this second phase of spindle orientation, as its maintenance at the apical cortex promotes perpendicular-directed telophase correction. Moreover, the fidelity of telophase correction relies on the actin-scaffolding α-E-catenin/vinculin/afadin pathway, highlighting a role for cell adhesion and cytoskeletal dynamics in division orientation. In this way, our findings now provide a mechanistic explanation for the randomized division orientation observed in *Ctnna1* mutants more than a decade ago (*Lechler and Fuchs, 2005*). Importantly, while our data support a model wherein vinculin regulates dynamic assembly of AJs, we cannot exclude the possibility that vinculin may play similar roles in cell-matrix integrin adhesions, which may also impact telophase correction.

## Corrective mechanisms in oriented cell divisions

Our findings contribute to a growing number of corrective mechanisms which can counterbalance stem cell division orientation errors in order to preserve tissue homeostasis. In *Drosophila* neuroblasts, the "telophase rescue" pathway—mediated by the scaffolding protein Dlg and motor protein Khc73—can compensate for errors in spindle orientation by relocalizing fate determinants, thus preserving normal daughter cell fates (*Cai et al., 2001*; *Peng et al., 2000*; *Siegrist and Doe, 2005*). However, telophase rescue differs from the telophase correction we report here in that division orientation errors are not corrected in telophase, but rather, the fate determinants themselves are repositioned relative to the new division axis. In the developing epidermis, it has been shown that Insc overexpression can promote apical LGN localization and drive an increase of perpendicular divisions (*Poulson and Lechler, 2010*; *Williams et al., 2011*), but that under some circumstances, NuMA can redistribute laterally, perhaps in an effort to prevent the hyper-differentiation that would be driven by excessive asymmetric divisions (*Poulson and Lechler, 2010*). Our data here, where afadin, α-E-catenin, and vinculin can override the perpendicular-correcting cue provided by LGN, provide a potential molecular explanation for this plasticity.

Other examples of dynamic oriented cell divisions include cyst stem cells of the *Drosophila* testis, which display randomized spindle angles until anaphase, at which point one centrosome becomes anchored at the interface with the niche-defining hub cell, driving division away from the niche (*Cheng et al., 2011*). In addition, dividing cells within the monolayered *Drosophila* follicular epidermis partially extrude during mitosis and frequently demonstrate oblique division angles, which are corrected by reinsertion into the epithelium in an adhesion-dependent manner (*Bergstralh et al., 2015*). A more extreme example of this extrusion/reinsertion model has been observed in intestinal organoids, where mitotic intestinal stem cells migrate to the luminal surface and undergo planar divisions before reinserting into the epithelium on either side of a Paneth cell (*McKinley et al., 2018*). Furthermore, genetic alterations in MDCK cells—specifically, *Gpsm2* knockdown or *Par1b* overexpression—can drive out-of-plane divisions which are capable of correcting during anaphase via an apical actomyosin compressive force (*Lázaro-Diéguez and Müsch, 2017*; *Zheng et al., 2010*). Taken together, these studies and ours suggest that many of these corrective mechanisms rely on polarity, cell-adhesion, and actin dynamics.

## Insights into epidermal cell fate specification

In the *Drosophila* neuroblast, the orientation of cell division is directly linked to cell fates via the asymmetric inheritance of transcription factors and other fate determinants which promote differentiation in one daughter cell and preserve stemness in the other (*Bergstralh et al., 2017*; *Knoblich, 2008*). While no such fate determinant has been identified in epidermal progenitors, our results add to a growing body of evidence that division orientation and cell fates are tightly linked. While previous studies have used short-term lineage tracing to correlate patterns of division orientation with fate choices (*Poulson and Lechler, 2010*; *Williams et al., 2014*), the lineage tracing experiments

**Figure 9.** Two-step model of division orientation. Model of OCD in the embryonic epidermis. During stratification, LGN (green) is recruited to the apical cortex in ~50% of mitoses, promoting perpendicular divisions. For OCDs with perpendicular and planar anaphase orientations, the division angle is fixed at anaphase onset, exhibiting minimal change in radial orientation during telophase. Importantly, the activity of LGN and its binding partners is imprecise, frequently resulting in oblique orientations at anaphase. In these cases, the apical daughter either retains or loses basement membrane contact following cytokinesis (red or blue nuclei, respectively). If contact is maintained, the apical daughter will reorient into a planar position. In contrast, if contact is lost, the apical daughter further stacks above its basal partner. Upon loss of α-E-catenin, vinculin, or afadin, telophase reorientation in either direction fails, resulting in persistent oblique divisions. In comparison, LGN loss reduces perpendicular anaphase orientations, while oblique divisions are properly corrected in a contact dependent manner. *Afdn* loss on an *Gpsm2* mutant background restores oblique divisions and largely rescues the *Gpsm2* differentiation defect.

performed in this study are the first to demonstrate that perturbations to division orientation lead to altered cell fate outcomes.

Importantly, given the timing of telophase correction, our observations also shed new light on the timing and speculative mechanisms of cell fate commitment during mitosis. In the normal developing epidermis, a large proportion of mitoses (30–40%) progress to anaphase at oblique orientations. While telophase reorientation normally sorts these indeterminate divisions into symmetric or asymmetric outcomes, the evidence from loss-of-function experiments in *Ctnna1* and *Afdn* mutants—resulting in a hyper-stratified epidermis and lineages biased toward differentiation—suggests that a significant portion of oblique divisions are operationally asymmetric, likely resulting in differentiation of the obliquely-positioned daughter cell. Furthermore, these data suggest that retention of basement membrane contact is a potentially potent driver of basal progenitor identity. Taken together,

our results indicate that while the early presence of LGN in pro/metaphase may bias cells towards adopting an asymmetric outcome, the finality of this decision is not determined until telophase reorientation mechanisms push or pull cells into the suprabasal or basal layers, respectively. Telophase correction thus provides a potential source of plasticity in the fate choices made by epidermal basal cells. It is tempting to speculate that AJ components in mitotic cells function as a mechanosensor that tranduces information about the local cellular environment that favors planar correction when tension is high and perpendicular correction when tension is low.

# Materials and methods

## Key resources table

| Reagent type (species) or resource | Designation | Source or ref. | Identifiers | Additional information |
|---|---|---|---|---|
| Strain, strain background (*Mus musculus*) | mTmG | Jackson Labs | IMSR Cat# JAX:007576, RRID:IMSR_JAX:007576 | |
| Strain, strain background (*Mus musculus*) | Krt14*Cre* | (*Dassule et al., 2000*) | | |
| Strain, strain background (*Mus musculus*) | Krt14*H2B-GFP* | (*Tumbar et al., 2004*) | | |
| Strain, strain background (*Mus musculus*) | CD1 | Charles River | IMSR Cat# CRL:022, RRID:IMSR_CRL:022 | New females integrated into colony every ~ 5 years to maintain as outbred strain. |
| Strain, strain background (*Mus musculus*) | LSL-Confetti | Jackson Labs | IMSR Cat# JAX:013731, RRID:IMSR_JAX:013731 | |
| Strain, strain background (*Mus musculus*) | Krt14*CreER* | Jackson Labs | IMSR Cat# JAX:005107, RRID:IMSR_JAX:005107 | |
| Strain, strain background (*Mus musculus*) | Afdn*fl/fl* | (*Beaudoin et al., 2012*) | | |
| Antibody | Guinea-pig polyclonal anti-LGN | (*Williams et al., 2011*) | | (1:500 dilution) |
| Antibody | Rabbit monoclonal anti-survivin (71G4B7) | Cell Signaling | Cat# 2808, RRID:AB_2063948 | (1:1000 dilution) |
| Antibody | Chicken polyclonal anti-GFP | Abcam | Cat# ab13970, RRID:AB_300798 | (1:2000 dilution) |
| Antibody | Rat monoclonal anti-mCherry (16D7) | Thermo Fisher Scientific | Cat# M11217, RRID:AB_2536611 | (1:2000 dilution) |
| Antibody | Rabbit polyclonal anti-LGN | Millipore | Cat# ABT174 | (1:2000 dilution) |
| Antibody | Rat monoclonal anti-β4 integrin | Thermo-Fisher | BD Biosciences Cat# 553745, RRID:AB_395027 | (1:1000 dilution) |
| Antibody | Rabbit polyclonal anti-Gai3 | EMD Millipore | Millipore Cat# 371726–50 UL, RRID:AB_211897 | (1:500 dilution) |
| Antibody | Goat polyclonal anti-dynactin | Abcam | Abcam Cat# ab11806, RRID:AB_298590 | (1:500 dilution) |
| Antibody | Mouse (IgM) monoclonal anti-NuMA | BD Transduction Labs | BD Biosciences Cat# 610562, RRID:AB_397914 | (1:300 dilution; use Jackson labs Donkey anti-IgM Cy3 conjugated secondary) |

*Continued on next page*

*Continued*

| Reagent type (species) or resource | Designation | Source or ref. | Identifiers | Additional information |
|---|---|---|---|---|
| Antibody | Rabbit polyclonal anti-histone H3, phospho (Ser10) | Millipore | Cat# 06–570, RRID:AB_310177 | (1:1000 dilution) |
| Antibody | Rat monoclonal anti-a-tubulin | EMD Millipore | Millipore Cat# CBL270, RRID:AB_93477 | (1:500 dilution) |
| Antibody | Rabbit polyclonal anti-pericentrin | Covance | Covance Cat# PRB-432C-200, RRID:AB_291635 | (1:500 dilution) |
| Antibody | Rabbit polyclonal anti-cytokeratin 10 (Poly19054) | Bio-Legend | Cat# 905404, RRID:AB_2616955 | (1:1000 dilution) |
| Antibody | Rabbit polyclonal anti-Par3 | EMD Millipore | Millipore Cat# 07–330, RRID:AB_2101325 | (1:500 dilution) |
| Antibody | Rat monoclonal anti-E-cadherin (ECCD-2) | Life Technologies | Thermo Fisher Scientific Cat# 13–1900, RRID:AB_2533005 | (1:1000 dilution) |
| Antibody | Goat polyclonal anti-E-cadherin | R and D systems | R and D Systems Cat# AF748, RRID:AB_355568 | (1:1000 dilution) |
| Antibody | Rabbit polyclonal anti-a-E-catenin | Invitrogen | Thermo Fisher Scientific Cat# 71–1200, RRID:AB_2533974 | (1:300 dilution; tissue sections) |
| Antibody | Rat monoclonal anti-a18 | (*Yonemura et al., 2010*) | | (1:10000 dilution; tissue sections) (1:2000 dilution; wholemounts) |
| Antibody | Rabbit polyclonal anti-vinculin | Gift from Dr. Keith Burridge | | (1:1000 dilution) |
| Antibody | Mouse monoclonal anti-vinculin | Sigma Aldrich | Sigma-Aldrich Cat# V9131, RRID:AB_477629 | (1:500 dilution) |
| Antibody | Rabbit polyclonal anti-afadin | Sigma Aldrich | Sigma-Aldrich Cat# A0224, RRID:AB_257871 | (1:500 dilution) |
| Antibody | Mouse monoclonal anti-pMLC2 (Ser19) | Cell Signaling | Cell Signaling Technology Cat# 3675, RRID:AB_2250969 | (1:500 dilution) |
| Chemical compound | Phalloidin AF-647 conjugated | Life Technologies | Thermo Fisher Scientific Cat# A22287, RRID:AB_2620155 | (1:500 dilution) |
| Cell line(s) (*Mus musculus*) | Primary keratinocytes | This publication. | | Isolated as described in Materials and methods section. |
| Chemical compound | Tamoxifen | Sigma-Aldrich | Cat# T5648 | |
| Software | FIJI | Source: https://imagej.net/Fiji Reference: DOI: 10.1038/nmeth.2019 | | |

## Animals

Mice were housed in an AAALAC-accredited (#329; June 2017), USDA registered (55-R-0004), NIH welfare-assured (D16-00256 (A3410-01)) animal facility. All procedures were performed under IACUC-approved animal protocols (16-162). For live imaging experiments we utilized either: (1) mT/mG (*Gt(ROSA)26Sor*$^{tm4(ACTB-tdTomato,-EGFP)Luo}$/J; Jackson Labs #007576 via Liqun Luo, Stanford University) homozygous females with at least one copy of the *Krt14*$^{Cre}$ allele (*Dassule et al., 2000*) (crossed to males of the identical genotype), or (2) *Krt14*$^{H2B-GFP}$ (*Tumbar et al., 2004*) and *Rosa26*$^{mT/mG}$ heterozygous females (crossed to identical males). For lineage tracing experiments (see below for additional details) we crossed *Krt14*$^{CreER}$; *Rosa26*$^{Confetti}$ females to identical males (Tg

(KRT14-cre/ERT)20Efu; Jackson Labs #005107/Gt(ROSA)26Sor$^{tm1(CAG-Brainbow2.1)Cle}$; Jackson Labs #013731). For fixed sample imaging, wild-type CD1 mice (Charles River; #022) were utilized. *Afdn$^{fl/fl}$* animals (*Beaudoin et al., 2012*) were maintained on a mixed C57B6/J CD1 background and either bred to the same *Krt14-Cre* allele or injected with lentiviral Cre-mRFP1 (see below). The procedure for producing, concentrating and injecting lentivirus into amniotic fluid of E9.5 embryos has been previously described and is briefly detailed below (*Beronja et al., 2010*).

## Live imaging

The live imaging protocol used in this study was adapted from the technique recently described by the Devenport lab (*Cetera et al., 2018*). A 1% agar solution/media solution containing F-media (3:1 DMEM:F12 + 10% FBS + 1% Sodium bicarbonate + 1% Sodium Pyruvate + 1% Pen/Strep/L-glut mix), was cooled and cut into 35mm discs. Epidermal samples measuring ~4-6mm along the AP axis and ~2-3mm along the medial-lateral axis were extracted from the mid-back of E16.5 mT/mG embryos. These explants were placed dermal-side down onto the gel/media disc, then sandwiched between the gas-permeable membrane of a 35mm lumox culture dish (Sardstedt; 94.6077.331). Confocal imaging was performed utilizing a Zeiss LSM 710 Spectral confocal laser scanning microscope equipped with a 40X/1.3 NA Oil Plan Neo objective. Images were acquired with 5 minute intervals and a Z-series with 0.5 μm step-size (total depth ranging from 20-30 microns) for 3-9 hours. Explants were cultured at 37°C with 5.0% $CO_2$ for >1.5 hours prior to- and throughout the course of imaging. Divisions occurring close to the tissue edge or showing any signs of disorganization/damage were avoided to exclude morphological changes associated with wound-repair. 4D image sets were deconvolved using AutoQuant X3 and processed using ImageJ (Fiji).

## Lentiviral injections

For full protocol, please see Beronja, et al. (ref. 24). This protocol is approved via IACUC #16-162/ 19-155. Pregnant CD1, *mTmG/Krt14-Cre*, or *Afdn$^{fl/fl}$* females were anesthetized and the uterine horn pulled into a PBS filled dish to expose the E9.5 embryos. Embryos and custom glass needles were visualized by ultrasound (Vevo 2100) to guide microinjection of ~0.7 μl of concentrated lentivirus into the amniotic space. Three to ten embryos were injected depending on viability and litter size. Following injection, the uterine horn(s) were reinserted into the mother's thoracic cavity, which was sutured closed. The incision in the skin was resealed with surgical staples and the mother provided subcutaneous analgesics (5 mg/kg meloxicam and 1-4 mg/kg bupivacaine). Once awake and freely moving, the mother was returned to its housing facility for 5-7 days, at which point E14.5-16.5 embryos were harvested and processed accordingly.

## Lineage tracing

Krt14$^{CreER}$; Rosa26$^{Confetti}$ females were mated to males with the identical genotype. At E9.5, ~half of the viable embryos were injected with *Afdn$^{2711}$* H2B-mRFP1 high titer lentivirus (see above for detailed surgical procedure). Activation of the *Krt14$^{CreER}$* allele was initiated by tamoxifen (dosed at 100 μg per gram dam mass) delivered by oral gavage at E14.5, five days following lentiviral injection). Females were monitored for 24 hours following tamoxifen dosing for signs of abortion or distress. Embryos were harvested at E17.5 (~72 hours after tamoxifen delivery) and backskins were embedded in OCT and sectioned sagittally (8μm thick sections). Slides were stained with Abcam Chicken αGFP polyclonal antibody (Abcam ab13970) which enhanced the membrane-CFP, nuclear-GFP, and cytoplasmic-YFP fluorophores of the *Confetti* allele. Images were acquired for every labeled clone using a 40x/1.15NA objective with a 1.5X digital zoom. Sparse clones (<1% total cells) were evaluated for both the number of basal and suprabasal cells (distinguished by staining with αKrt10 antibody; *Figure 8I*). Clones with only suprabasal cells in the stratum spinosum or first stratum granulosum (SG3) layer were assumed to be delamination events – those above SG3 were excluded. Suprabasal (SB) to basal cell ratios were quantified for each clone by dividing the # of SB cells by the # of basal cells. Clones with a ratio >1 were binned as 'SB-rich' while clones with a ratio <1 were binned as 'basal-rich' – clones with an equal number (ratio = 1) were binned as 'balanced'.

## Constructs and RNAi

For afadin and vinculin RNAi targeting, we tested ~10 shRNAs for knockdown efficiency in primary keratinocytes. These sequences were selected from The RNAi Consortium (TRC) Mission shRNA library (Sigma) versions 1.0, 1.5, and 2.0 and cloned using complementary annealed oligonucleotides with AgeI/EcoRI linkers. For LGN and α-catenin, we utilized an shRNA that had been previously validated with our lentiviral injection technique (*Beronja et al., 2010*; *Williams et al., 2011*). shRNA clones are identified by the gene name with the nucleotide base (NCBI Accession number) where the 21-nucleotide target sequence begins in superscript (e.g. $Afdn^{2711}$). Lentivirus was packaged in 293FT or TN cells using the pMD2.G and psPAX2 helper plasmids (Addgene plasmids #12259 and #12260, respectively). For knockdown screening, primary keratinocytes were seeded at a density of ~150,000 cells per well into 6-well plates and grown to ~80% confluency in E-Low calcium medium and infected with an MOI of ~1. Approximately 48 h post-infection, keratinocytes were treated with puromycin (2 μg/mL) to generate stable cell lines. After 3-4 days of puromycin selection, cells were lysed and RNA isolated using the RNeasy Mini Kit (Qiagen). cDNA was generated and amplified from 10-200 μg total RNA using either Superscript VILO (Invitrogen) or iScript (Bio-Rad). mRNA knockdown was determined by RT-qPCR (Applied Biosystems 7500 Fast RT-PCR) using 2 independent primer sets for each transcript with *Hprt1* and cyclophilin B (*Ppib2*) as reference genes and cDNA from stable cell lines expressing Scramble shRNA as a reference control. Primer efficiencies were determined using dose-response curves and required to be >1.8, with relative transcript abundance determined by the $\Delta\Delta CT$ method. RT-qPCR runs were performed in triplicate with the mean knockdown efficiency determined by calculating the geometric mean of the $\Delta\Delta CT$ values for at least two independent technical replicates. The following primer sequences were used: *Afdn* (fwd-1: 5'-ACGCCATTCCTGCCAAGAAG -3', rev-1: 5'- GCAAAGTCTGCGGTATCGGTAGTA -3'; fwd-2: 5'-GGGGATGACAGGCTGATGAAA -3', rev-2: 5'- CGATGCCGCTCAAGTTGGTA -3'), *Vcl* (fwd-1: 5'-TACCAAGCGGGCACTTATTCAGT -3', rev-1: 5'- TTGGTCCGGCCCAGCATA -3'; fwd-2: 5'- AAGGC TGTGGCTGGAAACATCT -3', rev-2: 5'- GGCGGCCATCATCATTGG -3'). The following shRNA targeting sequences were used: $Afdn^{2711}$ (5'- CCTGATGACATTCCAAATATA -3'), $Vcl^{3466}$ (5'- CCCTG TACTTTCAGTTACTAT -3'), $Vcl^{2803}$ (5'- CCACGATGAAGCTCGGAAATG -3'), $Ctnna1^{912}$ (5'-CGCTC TCAACAACTTTGATAA -3'), $Gpsm2^{1617}$ (5'- GCCGAATTGGAACAGTGAAAT -3'), Scramble (5'-CAACAAGATGAAGAGCACCAA-3').

## Antibodies, immunohistochemistry, and fixed imaging

E14.5 embryos were mounted whole in OCT (Tissue Tek) and frozen fresh at -20°C. E16.5 embryos were skinned and flat-mounted on Whatman paper. In both cases, infected and uninfected littermate controls were mounted in the same blocks to allow for direct comparisons on the same slide. For α-tubulin staining of metaphase spindles, samples were kept warm and pre-fixed with room-temperature 4% paraformaldehyde for 10 minutes before OCT embedding. Frozen samples were sectioned (8 μm thick) on a Leica CM1950 cryostat, mounted on SuperFrost Plus slides (ThermoFisher) and stored at -80°C. For staining, sections were thawed at 37°C for 5-15 min, fixed for 5 min with 4% paraformaldehyde, washed with PBS and blocked for 1h with gelatin block (5% NDS, 3% BSA, 8% cold-water fish gelatin, 0.05% Triton X-100 in PBS). Primary antibodies were diluted in gelatin block and incubated overnight in a humidity chamber at 4°C. Slides were then washed with PBS and incubated with secondary antibodies diluted in gelatin block at room temperature (~25°C) for 2 hours, counterstained with DAPI (1:2000) for 5 minutes and mounted in ProLong Gold (Invitrogen). Actin was visualized by phalloidin-AF647 staining (Life Technologies; 1:500) simultaneously with secondary antibody incubation. Images were acquired using LAS AF software on a Leica TCS SPE-II 4 laser confocal system on a DM5500 microscope with ACS Apochromat 20x/0.60 multi-immersion, ACS Apochromat 40x/1.15 oil, or ACS Apochromat 63x/1.30 oil objectives.

The following primary antibodies were used: survivin (rabbit, Cell Signaling 2808S, 1:500), LGN (*Williams et al., 2011*) (guinea pig, 1:500), LGN (rabbit, Millipore ABT174, 1:2000), phospho-histone H3 (rat, Abcam ab10543, 1:1,000), mCherry (rat, Life Technologies M11217, 1:1000-3000), β4-integrin (rat, ThermoFisher 553745, 1:1,000), Gαi3 (rabbit, EMD Millipore 371726, 1:500), GFP (chicken, Abcam ab13970, 1:1,000), dynactin (goat, Abcam ab11806, 1:500), NuMA (mouse IgM, BD Transduction Labs 610562, 1:300), α-tubulin (rat, EMD Millipore CBL270, 1:500), pericentrin (rabbit, Covance PRB-432C, 1:500), Par3 (rabbit, EMD Millipore 07-330, 1:500), E-cadherin (rat, Life Technologies

131900, 1:1,000), E-cadherin (goat, R&D System AF748, 1:1,000), α-E-catenin (rabbit, Invitrogen 71-1200, 1:300) α18 (rat, generous gift of Dr. Nagafuchi at Nara Medical University, 1:10,000), vinculin (mouse IgG, Sigma V9131, 1:500), vinculin (rabbit, generous gift of Dr. Keith Burridge at University of North Carolina, 1:1000), afadin (rabbit, Sigma A0224, 1:500), pMLC2 (Ser19) (mouse IgG, Cell Signaling 3675S, 1:1000). Actin labeling achieved via Phalloidin-AF647 (Life Technologies A22287, 1:500) in secondary antibodies

The following secondary antibodies were used (all antibodies produced in donkey): anti-rabbit AlexaFluor 488 (Life Technologies, 1:1000), anti-rabbit Rhodamine Red-X (Jackson Labs, 1:500), anti-rabbit Cy5 (Jackson Labs, 1:400), anti-rat AlexaFluor 488 (Life Technologies, 1:1000), anti-rat Rhodamine Red-X (Jackson Labs, 1:500), anti-rat Cy5 (Jackson Labs, 1:400), anti-guinea pig AlexaFluor 488 (Life Technologies, 1:1000), anti-guinea pig Rhodamine Red-X (Jackson Labs, 1:500), anti-guinea pig Cy5 (Jackson Labs, 1:400), anti-goat AlexaFluor 488 (Life Technologies, 1:1000), anti-goat Cy5 (Jackson Labs, 1:400), anti-mouse IgG AlexaFluor 488 (Life Technologies, 1:1000), anti-mouse IgG Cy5 (Jackson Labs, 1:400), anti-mouse IgM Cy3 (Jackson Labs, 1:500).

## Keratinocyte culture and Calcium-shift assays

Primary mouse keratinocytes were maintained in E medium with 15% chelated FBS and 50 µM $CaCl_2$ (E low medium). For viral infection, keratinocytes were plated at ~150,000 cells per well in a 6-well plate and incubated with lentivirus in the presence of polybrene (1 µg/mL) and centrifuged at 1,100 x*g* for 30 min at 37°C. All shRNA cell lines were derived from the same wild-type lineage (primary CD1 mouse keratinocytes isolated from P3 backskins). Stable cell lines were generated/maintained by adding puromycin (2 µg/mL) 48 h after infection and continual antibiotic treatment following. The *Afdn*$^{fl/fl}$ and *Afdn*$^{fl/fl}$; *Krt14*$^{Cre}$ (*Afdn*-cKO) keratinocyte lines were isolated from P3 littermates and used at low passage (<P10). Cell line identity was doubly confirmed by knockdown/knockout specificity via immunofluorescent staining. All lines tested negative for mycoplasma using the ATCC 30-1012K kit. Calcium shifts were performed by seeding ~45,000 low passage cells (<P10) per well into 8-well Permanox chamber slides (Lab-Tek 177445) coated with poly-L-lysine, collagen, and fibronectin. Once cells reached ~85% confluency (~12-16 hours) cells were switch to high $Ca^{2+}$ (1.5mM) medium and grown for the indicated period of time (30 min to 8 hours). Cells were fixed with 4% paraformaldehyde in PBS warmed to room-temperature. Immunostaining was performed using the same protocol as for tissue sections (see above).

3T3 fibroblasts and HEK-293 cells – both of which are included on the International Cell Line Authentication Committee's register of misidentified cell lines (version 9) were specifically used in primary keratinocyte isolation and lentiviral production, respectively. Neither cell line was utilized in any experimental procedures.

## Measurements, quantification, graphing, and statistics

### Spindle and division orientation

Mitotic cells in metaphase were identified based on nuclear morphology. Metaphase spindle orientation was measured as the angle between a vector orthogonal to the metaphase plate and parallel to the basement membrane. Anaphase cells were identified by both nuclear condensation and widely distributed surviving staining between daughter cells. Telophase cells were distinguished due to reduced nuclear condensation and dual-punctate Survivin staining. Division orientation was measured as the angle between a vector connecting the center of each daughter nucleus and a vector running parallel to the basement membrane. The same methodology was used to measure division orientation in live imaging experiments. In cases lacking nuclear labeling, the position of the nuclei was inferred based on cell volume/shape changes. Telophase correction (θ-φ) was quantified as the difference between division orientation at anaphase onset (φ) and division orientation 1 h later (θ). The presence of basal contact for the more apical daughter was determined by analyzing cell morphology in both *en face* and orthogonal perspectives.

### Adhesion assays

Quantification of fluorescence intensity in adhesion assays was performed by orthogonal linescans at three positions along the junction length (~25th, 50th, and 75th quarter). In cases where junctions appeared punctate, discreet puncta were evaluated to avoid measuring regions lacking junction

formation. Signal centers were set based on maximum intensity of either E-cadherin or α-E-catenin (where appropriate). To quantify ratios, the geometric mean fluorescent intensities of the 3 values nearest the junction center were used. Quantification of junction continuity was performed by line-scans of E-cadherin fluorescence intensity along the entire length of the junction, excluding the vertex of multiple cells (i.e. tricellular junctions). We then calculated % of these intensity measurements above a threshold, which was evaluated for each individual junction using the mean center intensity of three orthogonal scans described earlier in this paragraph.

### LGN localization/intensity

LGN localization patters (e.g. apical, weak/absent, or other) were determined for cells labeled with pHH3, irrespective of the lentiviral H2B-RFP reporter to avoid bias. Imaging was performed with WT controls and experimental samples on the same slide to avoid variation in antibody staining. Radial localization of LGN was measured by determining the angle between two vectors: one drawn from the LGN signal center to the center of the nucleus, the other drawn parallel to the basement membrane. Crescents oriented at the apical side were given positive values, while those at the basal side were given negative values. Radial variance between LGN signal and spindle or division axis were determined by drawing two vectors: one for the radial orientation of the LGN signal center and a second between either the spindle poles or between the center of the daughter nuclei. Radial fluorescent intensity values were measured by linescans originating at the site of basement membrane contact and tracing the edge of the cell. Each measurement along the length of the scan was then set as a part of whole, operating with the assumption that ~50% of the total length would represent the apical surface.

### Whole mount fluorescence intensity

Quantification of fluorescence intensity in wholemount imaging of oblique telophase cells (as performed for pMLC2, phalloidin, α18 and α-E-catenin in *Figure 3*) was performed by drawing linescans in the en-face perspective around the entire cortex at a series of focal planes: apical plane of the apical daughter, 'endfoot' of the apical daughter, mid-cell of the basal daughter and mid-cell of an interphase basal neighboring cell. For each image, background levels were determined using mean fluorescent intensity for each channel in a neighboring cell nucleus. This background value was subtracted from the mean cortical intensity of the appropriate channel. Fluorescence intensity ratios were quantified using background subtracted mean fluorescence intensities.

Cell density and live cell division orientation. Local cell density was calculated by counting the number of neighboring cells and dividing this number by their area. Area was measured in a single z-plane determined to be the cell center by orthogonal slices. Regions where the tissue sloped at an extreme angle were excluded due to inability to capture cell centers for all neighbors.

### Differentiation/stratification analyses

E14.5 differentiation was quantified by imaging ~10 regions of the backskin in sagittal sections stained with β4 integrin, K10 and H2B-RFP. For each region, the number of basal and suprabasal cells were counted and the length of the region measured by the length of the underlying basement membrane (to account for tissue wrinkling/curvature). To quantify cell density, cell counts were divided by the length of each region in microns. At later stages (E16.5 or E18.5) we quantified K10 thickness by imaging ~10 regions of the backskin in sagittal sections stained with β4 integrin, K10 and H2B-RFP. Using the K10 channel, a thresholded, binary mask was created and filled, then used to measure the area above threshold. This thresholded area was then divided by the length of the underlying basement membrane (measured by β 4 integrin stain). *n* values for these analyses are representative of the number of regions imaged.

### Mitotic index

Mitotic Index was quantified at E14.5 and E16.5 by imaging the entire length of sagittally-sectioned backskin stained for β4 and pHH3. pHH3+ basal cells were counted and the length of the entire backskin was measured by length of the underlying basement membrane. *n* values are indicative of the number of individual embryos analyzed.

All statistical analyses and graphs were generated using GraphPad Prism 8 and Origin 2015 (OriginLab). Error bars represent standard error of the mean (s.e.m.) unless otherwise noted. Statistical tests of significance were determined by Mann-Whitney $U$-test (non-parametric) or student's t-test (parametric) depending on whether the data fit a standard distribution (determined by pass/fail of majority of the following: Anderson-Darling, D'Agostino & Pearson, Shapiro-Wilk, and Kolmogorov-Smirnov tests). Cumulative frequency distributions were evaluated for significant differences by Kolmogorov-Smirnov test. $\chi^2$ tests were utilized to evaluate expected (control) against experimental distributions of categorical values (e.g. LGN apical/absent/other distributions). All box-and-whisker plots are displayed as Tukey plots where the box represents the interquartile range (IQR, $25^{th}$-$75^{th}$ percentiles) and the horizontal line represents the median. Whiskers represent 1.5x IQR unless this is greater than the min or max value. Figures were assembled using Adobe Photoshop and Illustrator CC 2017.

## Acknowledgements

We thank members of the Williams and Peifer Labs for their critical feedback. We thank Dr. Danelle Devenport (Princeton) for graciously sharing her live imaging protocol. We thank Dr. Akira Nagafuchi (Nara Medical University) for sharing the α18 antibody. We thank Dr. Brent Hoffman and Evan Gates (Duke University) for their effort and feedback regarding E-cadherin mediated junctional tension. We thank Dr. Keith Burridge (UNC) for sharing the vinculin (Rb) polyclonal antibody. We thank Kendra Niederkorn for design input into the model in Figure 9. KJL was supported by an NIH Ruth L Kirschstein Predoctoral National Research Service Award (F31 DE026956). KMB is supported by an NIH/NIDCR K08 Mentored Clinical Scientist Research Career Development Award (DE026537). SW was supported by a Sidney Kimmel Scholar Award (SKF-15-165).

## Additional information

### Funding

| Funder | Grant reference number | Author |
| --- | --- | --- |
| National Institute of Dental and Craniofacial Research | Predoctoral Fellowship F31 DE026956 | Kendall J Lough |
| National Institute of Dental and Craniofacial Research | Career Development Award K08 DE026537 | Kevin M Byrd |
| Sidney Kimmel Foundation for Cancer Research | Scholar Award SKF-15-165 | Scott E Williams |

The funders had no role in study design, data collection and interpretation, or the decision to submit the work for publication.

### Author contributions

Kendall J Lough, Conceptualization, Data curation, Formal analysis, Supervision, Funding acquisition, Validation, Investigation, Visualization, Methodology; Kevin M Byrd, Conceptualization, Investigation, Methodology; Carlos P Descovich, Data curation, Formal analysis, Methodology; Danielle C Spitzer, Abby J Bergman, Data curation, Investigation; Gerard MJ Beaudoin III, Louis F Reichardt, Resources; Scott E Williams, Conceptualization, Resources, Supervision, Investigation, Methodology, Project administration

### Author ORCIDs

Kendall J Lough https://orcid.org/0000-0001-9663-6983
Kevin M Byrd http://orcid.org/0000-0002-5565-0524
Carlos P Descovich http://orcid.org/0000-0002-6366-5195
Danielle C Spitzer https://orcid.org/0000-0003-4827-1857
Scott E Williams https://orcid.org/0000-0001-9975-7334

### Ethics

Animal experimentation: Mice were housed in an AAALAC-accredited (#329; June 2017), USDA registered (55-R-0004), NIH welfare-assured (D16-00256 (A3410-01)) animal facility. All procedures were performed under IACUC-approved animal protocols (19-155).

### Decision letter and Author response

Decision letter https://doi.org/10.7554/eLife.49249.sa1
Author response https://doi.org/10.7554/eLife.49249.sa2

## Additional files

### Supplementary files

• Transparent reporting form

### Data availability

All data generated or analyzed during this study are included in the manuscript and supporting files. Source data files have been provided for all Figures.

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
