## [Decision Letter]

**Acceptance summary:**

This manuscript reports and characterizes an interesting mechanism to regulate the orientation of cell division within epithelial tissues. Surprisingly, the authors find that additional mechanisms correct the orientation of a dividing cell in the later transition to telophase rather than the more well-studied anaphase. Interestingly, the authors implicate adherens junction proteins in this phase of spindle orientation. This work will be of interest to scientists in multiple fields including cellular adhesion, stem cell biology, and epithelial biology.

**Decision letter after peer review:**

Thank you for submitting your article "Telophase correction refines division orientation in stratified epithelia" for consideration by *eLife*. Your article has been reviewed by three peer reviewers, and the evaluation has been overseen by a Reviewing Editor and Anna Akhmanova as the Senior Editor. The following individuals involved in review of your submission have agreed to reveal their identity: Alpha Yap (Reviewer #1); Carien M Niessen (Reviewer #3).

The reviewers have discussed the reviews with one another and the Reviewing Editor has drafted this decision to help you prepare a revised submission.

Summary:

This manuscript reports and characterizes an interesting mechanism to regulate the orientation of cell division. Whereas much of this field has concentrated on how spindle orientation may be defined at the onset of anaphase, the authors show that there are additional mechanisms that correct orientation of the dividing cell in the later transition to telophase. This is prompted by the observation that cells are found with intermediate orientations at anaphase, which is corrected by telophase. The authors show that such correction involves two mechanisms: 1) A role for LGN in "perpendicular" correction, that extends its current role in anaphase specification; and 2) A role for the AJ cytoskeleton, that functionally involves α-catenin, vinculin and afadin. The latter is interesting because it appears to be dominant: although LGN-depleted cells can correct to planar orientation (perhaps because of this junctional input), correction appears to be fundamentally compromised (division planes are close-to randomized) when these AJ-actin linkers are depleted.

Characterization of this phenomenon of orientation correction is a valuable contribution to the field. This value is enhanced because it has been i) performed in a complex tissue model; and ii) in a stratified epithelium where decisions of perpendicular vs. planar division have consequences for cell fate (as the authors pursue in Figure 7).

Essential revisions:

1) The authors should provide more data regarding the physiological importance of telophase rescue. Is the cortex of either the dividing cell or its immediate neighbours altered in e.g. afadin and vinculin depletion? The authors have studied this in an in vitro model of junction formation, but one might suspect that the orientation correction mechanism involves generating some anisotropy in the cortex, e.g. concentration of Myosin II. It is difficult to discern the cytokinetic furrow in most videos and to clearly tell when abscission occurs. This is especially true for planar rescues, which could as likely be to cell respreading as cells exit mitosis. Along these lines, it would be nice to include some of the H2B-RFP videos in addition to the cell membrane videos that are included in the supplementary data section.

2) There is very little analysis of phenotypes resulting from loss of telophase rescue. Thus, the physiological significance of this process is questionable. Is the same precocious differentiation seen in vinculin mutant and acatenin mutants as well as afadin. Do they have later phenotypes that are consistent with spindle orientation functions? In addition, some of the work linking these proteins to junctional tension/organization comes only from cultured keratinocytes though the mutants are in hand. It would be ideal to see adherens junction/actin/tension markers in intact skin.

3) This work seems somewhat at odds with other reports in the field, including those from the PI. For example, LGN knockdown cells were reported to have almost entirely planar divisions, yet here they are reported to be quite random until telophase. Were those previous spindle orientation measurements all performed in telophase? And what about divisions orientation quantitations from multiple other groups that have reported more clear spindle orientations during anaphase?

4) Ultimately, one would like to know how to "order" the contributions of α-catenin, vinculin and afadin in this phenomenon. For example, at first glance it is slightly surprising that vinculin should decrease apparent tension across α-catenin, when the recruitment of vinculin to α-catenin is thought to be promoted by application of tension to the latter. This might reflect a contribution of vinculin to forces applied on integrin adhesion (which are secondarily transmitted to junctions) or simply the difficulty of parsing pathways that may not be linear. I mention this purely as a comment out of interest.

5) Revised discussion of the role vinculin in integrins. The authors don't directly address whether vinculin function at integrin or cadherin adhesions are required for telophase rescue. This should be directly stated.

The authors don't comment on whether vinculin knockdown (and other mutants) have a significant effect on perpendicular spindle telophase rescue. It appears from the data that these might affect perpendicular as well as parallel spindles. If so, this should be discussed.

[Editors' note: further revisions were requested prior to acceptance, as described below.]

Thank you for resubmitting your work entitled "Telophase correction refines division orientation in stratified epithelia" for further consideration by *eLife*. Your revised article has been evaluated by Anna Akhmanova as the Senior Editor, a Reviewing Editor and three reviewers.

The manuscript has been improved but there are some remaining issues that need to be addressed before acceptance, as outlined below:

The reviewers thought that the conclusions on tension regulation are too strong and should be tuned down in the text and legend titles. Once these revisions are complete, the manuscript will be appropriate for publication in *eLife*.

Reviewer #1:

I think the authors have reasonably responded to the questions raised in my earlier review. This is now a near-gigantic manuscript that reports an interesting phenomenon. Given its size and depth one can also quibble about details. But I think that the field would be better served by having the story out for consideration and replication.

Reviewer #2:

Substantial changes have been made, both experimentally and in writing, that strengthens the manuscript. I think it is largely suitable for publication.

I really wish that they would have included analysis of tissue thickness/premature differentiation of vinculin mutants, which would have strengthened the results (this data should be in hand).

I do have some reservations in inconsistency with the data with much other published work including the α-catenin knockdown data and the data on delamination at early stages of stratification. The Devenport lab does not see this with live imaging (and neither do we), although the Wickstrom group has. Since they performed imaging at E14.5, it would be great if they could state whether the rate of delamination they see in their videos validates their lineage tracing.

Reviewer #3:

This paper is improved and has addressed some of my major concerns. The lineage tracing data have helped to show that there is a likely direct consequence in terms of cell density in the different layers, even though if I understand Figure 7G and H correctly, this seems to have no direct effect on epidermal thickness or differentiation (time point of analysis is not mentioned though in the legends of these panels).

I do still feel that the authors conclusions on tension regulation are too strong as it is all based on staining (Figures 3 and 4) and should be tuned down in the text and legend titles as no experiment directly addresses the role of tension or hyper contractility. For example, the phosphomyosin staining is suggestive of increased contractility, but there is no direct evidence for this (laser ablation, etc.) and in addition the resolution in the images does not really allow one to make this very clear statement as in the legend title that a hyper contractile basal endfoot directs planar telophase correction. I realise that such experiments, especially in vivo are not trivial, but even in none of their in vitro experiments the authors directly test tension, but it is all based on staining. The authors should not misunderstand me, but the strength is the telophase correction pathway and the role of junctions in this pathway that is independent of LGN, but whether these are direct consequences of tension is beyond this manuscript and can be suggested and discussed but requires quite a bit more to show that convincingly.

Having said that, and if the authors are willing to tune those conclusions down, I think this is a very nice contribution to *eLife*.

---

## [Author Response]

Essential revisions:1) The authors should provide more data regarding the physiological importance of telophase rescue. Is the cortex of either the dividing cell or its immediate neighbours altered in e.g. afadin and vinculin depletion? The authors have studied this in an in vitro model of junction formation, but one might suspect that the orientation correction mechanism involves generating some anisotropy in the cortex, e.g. concentration of Myosin II. It is difficult to discern the cytokinetic furrow in most videos and to clearly tell when abscission occurs. This is especially true for planar rescues, which could as likely be to cell respreading as cells exit mitosis. Along these lines, it would be nice to include some of the H2B-RFP videos in addition to the cell membrane videos that are included in the supplementary data section.

We have addressed the reviewers’ comment in several ways:

We have analyzed anisotropy of phosphorylated (Ser19) myosin light chain 2 (pMLC2) within oblique telophase cells displaying a characteristic basal endfoot as observed in planar correction via live imaging. These analyses were performed by whole mount staining E16.5 epidermis with phalloidin and pMLC2 and quantifying fluorescence intensity in rare, obliquely oriented telophase divisions (Figure 3D). These measurements demonstrated a greater accumulation of pMLC2 in the basal endfoot than compared to the apical cortex of the “correcting” daughter cell, suggesting anisotropic contractility during telophase correction.

Further supporting this conclusion, we stained E16.5 epidermal whole mounts with both the a18 and a polyclonal α-E-catenin antibody to estimate the relative abundance of the open, “tensile” conformation of α-E-catenin in the same context. This quantification (performed similarly to the methodology for pMLC2) demonstrated that the a18 epitope stained with greater intensity in the basal endfoot compared to the apical cortex of the “correcting” daughter cell, while total a-Ecatenin levels were unchanged between these two regions (Figure 3F; Figure 3—figure supplement 1E). Furthermore, the a18: α-E-catenin fluorescence intensity ratio was highest in the basal endfoot when compared to other mitotic scenarios (Figure 3G). This suggests that the increased contractility in the endfoot changes the conformation of α-E-catenin, thereby justifying our pursuit of these molecules as candidate regulators of telophase correction.

Given the rarity of these oblique, anaphase/early telophase divisions in fixed tissue we are unable to make clear conclusions regarding the dynamics of these molecules through mitotic progression. Ultimately, elucidating these mechanisms will likely require live imaging of fluorescently labeled transgenes for α-E-catenin, afadin, vinculin, myosin, and/or actin. We attempted such an experiment utilizing a miRFP-670 conjugated LifeAct construct packaged in high-titer lentivirus, but were unable to complete imaging due to difficulty in either A) achieving a useable infection rate or B) inability to detect dim fluorescent signal from the construct on the available microscopes. We are unaware of anyone successfully performing live-imaging of lentivirally-expressed fluorescent transgenes, other than our own success with the extremely bright H2B-RFP reporter. We predict that many of these transgenes will be relatively lowly expressed, making fluorescence detection difficult in thick, stratified epidermis (E16.5). While this presents an exciting avenue of future research, approaching these experiments will require substantial technical optimization in terms of viral packaging, expression, and image collection which we consider beyond the scope of the current manuscript.

2) There is very little analysis of phenotypes resulting from loss of telophase rescue. Thus, the physiological significance of this process is questionable. Is the same precocious differentiation seen in vinculin mutant and acatenin mutants as well as afadin. Do they have later phenotypes that are consistent with spindle orientation functions? In addition, some of the work linking these proteins to junctional tension/organization comes only from cultured keratinocytes though the mutants are in hand. It would be ideal to see adherens junction/actin/tension markers in intact skin.

We have addressed the reviewers’ comment in several ways:

We have analyzed the precocious differentiation phenotype at E14.5 for *Ctnna1* knockdown samples (*Ctnna1^912^*), which present with a nearly identical phenotype to the *Afdn* knockdown, suggesting the precocious differentiation observed in these mutants is more widely attributable to those mutants which fail to undergo telophase correction.

We have similarly analyzed the effects on differentiation at E16.5 for the following genotypes: *Afdn^2711^, Afdn^fl/fl^ + lenti-Cre* and *Ctnna1^912^*. all of which display sustained hyperstratification. Similarly, the *Afdn^2711^, Afdn^fl/fl^ + lenti-Cre* increased thickness further persists through E18.5. These data suggest that failed telophase correction increases differentiation throughout epidermal stratification.

We have also investigated the potential impact of increased proliferation on the observed differentiation phenotypes. While previous reports from α-E-catenin knockout epidermis confirmed a hyperproliferative phenotype, the same study (Berojna and Fuchs, 2010; cited numerous times) concluded that *Ctnna1* knockout clones were at a net growth disadvantage due to increased apoptosis. Similarly, these analyses were restricted to BrdU incorporation assays, which, while frequently correlative with rates of cell division, do not directly address rates of mitosis and could possibly be due to lengthened S-phase. Here, we have quantified mitotic index by counting pHH3+ cells per mm of backskin and shown at E14.5 and E16.5, that neither loss of *Afdn* nor *Ctnna1* results in a significant increase in mitotic events.

Lastly, we have performed short-term genetic lineage tracing experiments for *Afdn* knockdown and WT littermates (Figure 8I-K; Figure 8—figure supplement 1D-G) to examine clonal growth during stratification (E14.5-E17.5). Using this assay we were able to conclude the following:

– Loss of afadin results in clones with a greater number of differentiated suprabasal cells with no significant change in overall clone size, suggesting a shift towards asymmetric cell divisions, likely due to failure of oblique divisions to undergo planar correction.

– Delamination – an alternative differentiation pathway to asymmetric cell divisions – is largely unaffected by *Afdn* loss. In our clonal analyses, we assumed that suprabasal clones lacking any basal cells were likely the result of differentiation by delamination (without cell division). Given this assumption, loss of afadin did not significantly alter the rate of delamination, further suggesting that the increased stratification in these mutants is due specifically to errors in telophase correction.

– As WT and afadin knockdown clones had a similar number of total labeled cells (Figure 8—figure supplement 1E) we were able to reaffirm that loss of afadin had no impact on net growth.

Ultimately, these data further support the conclusion that the precocious and sustained hyperstratification observed in both mutants is most likely due to failed telophase correction. These data form the centerpiece of a newly added Figure 8, which focuses on characterizating the consequences of failed telophase correction. Pertinent discussion of these results has been added to a subheader in the Results section titled “Telophase correction impacts cell fate decisions”.

3) This work seems somewhat at odds with other reports in the field, including those from the PI. For example, LGN knockdown cells were reported to have almost entirely planar divisions, yet here they are reported to be quite random until telophase. Were those previous spindle orientation measurements all performed in telophase?

We thank the reviewers for pointing out the apparent contradiction between this study and previous results. We realize we did not contextualize our findings as clearly as we could have, so we have expanded on this in the first paragraph of the Results section:

“Our previous studies reported a bimodal distribution of division angles at late stages of mitosis and randomized division angles during metaphase (Williams et al., 2011; Williams et al., 2014), while other groups have reported that spindle rotation occurs during prometaphase and is fixed to a bimodal distribution by late metaphase/early anaphase (Poulson and Lechler, 2010; Seldin et al., 2016). While these studies agree that spindle rotation occurs, they come to different conclusions about when and how the spindle axis becomes fixed to a bimodal distribution.”

We further explain the rationale for our chosen method of using Survivin to label late-stage mitotic cells in the subsection “Randomized division orientation persists into anaphase”.

To improve the rigor and reproducibility of our studies of metaphase/anaphase/telophase spindle orientation in fixed tissue, we included three separate commonly-used strains of mice (CD1, C57 and 129), and separately analyze each strain (Figure 1—figure supplement 1A). Since each strain showed similar patterns, we rule out potential strain differences as an explanation for any reported differences in anaphase division orientation.

And what about divisions orientation quantitations from multiple other groups that have reported more clear spindle orientations during anaphase?

While there are examples of spindle or division orientation measurements made at metaphase or anaphase in the epidermis, each varies slightly in method from ours reported here. These are discussed in detail below:

Lechler and Fuchs, 2005 (cited numerous times): In Figure 1A-E the authors present data which first reported the presence of symmetric-planar and asymmetric-perpendicular divisions in the embryonic epidermis. These experiments utilized either DAPI and tubulin (Figure 2A,B, E) or DAPI with a Centrin-GFP allele (Figure 2C-E) to measure division angles. However, these data to not differentiate between anaphase and telophase.

Poulson et al., 2010 (cited numerous times): In Figure 2C-E the authors present data measuring spindle angles in vivo using Centrin-GFP or NuMA-GFP alleles. At prometaphase (Figure 2C-D) they demonstrate nearly random spindle orientation but show a dramatic shift towards bipolar angles at metaphase (Figure 2E). However, it is important to note that the measurements were made using an overexpressed spindle orientation effector, NuMA-GFP which possibly impacts division orientation behavior.

Seldin et al., 2016 (now cited): In Figure 3J the authors present data measuring spindle angles using IF staining for pHH3 to ID anaphase cells in E17.5 interfollicular epidermis. While the authors of this study state measurements were restricted specifically to anaphase, they do not present example images, making the discrepancy difficult to explore. The described methods suggest that anaphase cells were identified by pHH3 staining, which *may* persist into telophase (data delineating pHH3 antibody specificity through distinct stages of mitosis in epidermal progenitors does not exist, and in our hands has proven to be variable between distinct antibody). These measurements were also performed on thin sagittal sections, limiting the authors ability to observe both planar and oblique division angles, which may frequently occur out of the section plane. Lastly, these measurements were taken at E17.5, one day later than the data presented in this manuscript, which may alter the timing of division plane commitment.

We are unaware of other instances of metaphase or anaphase measurements of spindle/division orientation in the embryonic epidermis and would welcome more specific citations if we have missed a relevant data set. However, the above listed studies do not present data specifically measuring division orientation at anaphase in E16.5 or E14.5 interfollicular epidermis. The data evaluating metaphase/anaphase division angles were either collected in a mutant context overexpressing a key spindle orientation gene, or at a different timepoint (E17.5) using different methods of quantitation. Similarly, these data likely underestimate the proportion of oblique and planar divisions due to the fact that these measurements were made in sagittal sections, which impact the ability to observe divisions which occur out of the plane of the section.

4) Ultimately, one would like to know how to "order" the contributions of α-catenin, vinculin and afadin in this phenomenon. For example, at first glance it is slightly surprising that vinculin should decrease apparent tension across α-catenin, when the recruitment of vinculin to α-catenin is thought to be promoted by application of tension to the latter. This might reflect a contribution of vinculin to forces applied on integrin adhesion (which are secondarily transmitted to junctions) or simply the difficulty of parsing pathways that may not be linear. I mention this purely as a comment out of interest.

The reviewers are correct to highlight that initial tension across α-E-catenin (inducing the “open” conformation) would likely be required for vinculin and afadin binding (based on the findings of Buckley, Nelson and Dunn, 2014; Science as well as Pokutta and Weiss, 2002). However, this model is not exclusive to the possibility that vinculin binding may stabilize the open conformation of α-E-catenin, which would result in a larger proportion of α-E-catenin molecules receiving the a18 label in a fixed stain. The same may be true for afadin as well. Our data suggest that these two proteins (vinculin and afadin) cooperatively promote the stabilization of α-E-catenin’s open, “tensile” conformation, which in turn promotes the formation of the afadin/α-E-catenin and vinculin/α-E-catenin interactions. One interesting observation that we make is that it is not so much that a18 levels decrease, but rather that total α-E-catenin levels increase upon *Vcl* or *Afdn* loss (Figure 4—figure supplement 1A; Figure 1—figure supplement 2G, H). This suggests there may be compensatory upregulation of α-E-catenin (and perhaps other AJ components) to counter the weakened AJs. This is discussed in the subsection “Telophase corrective basal contacts are hyper-contractile”.

We agree that our data are far from conclusive, but should note that there is a decade of literature addressing mechanistically how vinculin binding regulates the tension sensing capabilities of α-E-catenin, and very little work has been done on afadin. Although we lack the capabilities to tease their “order” of incorporation into the complex apart, we have analyzed the spindle orientation phenotype in double mutants of *Vcl* and *Afdn*, which resemble single mutants: “Examination of division orientation in single and double mutants revealed that vinculin loss did not exacerbate the *Afdn cKO* phenotype, suggesting that these proteins do not act additively in the context of division orientation (Figure 5I).” Our in vitro nascent AJ formation experiments lack the temporal resolution to identify a sequential recruitment into AJs, and we believe the biochemical analyses required for fully elucidate the underlying mechanisms of these binding events are beyond the scope of this current project. We would be happy to discuss the implications of our data regarding a molecular model of vinculin/afadin binding to α-E-catenin but feel any discussion beyond what is already included in the text would be highly speculative.

5) Revised discussion of the role vinculin in integrins. The authors don't directly address whether vinculin function at integrin or cadherin adhesions are required for telophase rescue. This should be directly stated.The authors don't comment on whether vinculin knockdown (and other mutants) have a significant effect on perpendicular spindle telophase rescue. It appears from the data that these might affect perpendicular as well as parallel spindles. If so, this should be discussed.

We have addressed the reviewers’ comments in several ways.

We have added relevant discussion of vinculin in integrin based adhesions to the text. At the end of the first paragraph in the Discussion we have added the following statement: “Importantly, while our data support a model wherein vinculin regulates dynamic assembly of AJ, we cannot exclude the possibility that vinculin may play similar roles in cell-matrix integrin adhesions, which may also impact telophase correction.” We believe this statement is justified due to the fact that loss of α-E-catenin or afadin, neither of which have been implicated in cell-matrix adhesion, produce errors in telophase correction. However, the reviewers are correct to point out that we cannot completely exclude the possibility that vinculin serves roles at both cell-cell and cell-matrix adhesions, due in part to the difficulties in adequately labeling/staining vinculin in epidermal basal cells.

We have added specific comments highlighting that vinculin, α-E-catenin, and afadin loss results in errors to telophase correction *in both planar and perpendicular directions*.

[Editors' note: further revisions were requested prior to acceptance, as described below.]

The reviewers thought that the conclusions on tension regulation are too strong and should be tuned down in the text and legend titles. Once these revisions are complete, the manuscript will be appropriate for publication in eLife.

We have made efforts throughout the text to temper our language with respect to the conclusions we draw about tension regulation. See specific examples in the response to reviewer #3, below.

Reviewer #2:[…]I really wish that they would have included analysis of tissue thickness/premature differentiation of vinculin mutants, which would have strengthened the results (this data should be in hand).

While we agree that the addition of analysis of *Vcl* mutants would have made a nice addition to Figure 8, unfortunately these were not data that we had “in hand.” It would have been necessary for us to generate additional *Vcl* mutants at E14.5, which would have delayed resubmission by at least another month. We did perform a cursory examination of E16.5 *Vcl* mutants and did not observe obvious differentiation defects. However, these embryos were more mosaic (thus containing less mutant tissue) than either of the other mutants. It should also be noted that even in *Ctnna1* and *Afdn* mutants, the differentiation defect is more subtle at this age than at E14.5.

I do have some reservations in inconsistency with the data with much other published work including the α-catenin knockdown data and the data on delamination at early stages of stratification. The Devenport lab does not see this with live imaging (and neither do we), although the Wickstrom group has. Since they performed imaging at E14.5, it would be great if they could state whether the rate of delamination they see in their videos validates their lineage tracing.

I believe the only inconsistency between our findings and other published work regarding α-E-catenin is related to the expression of LGN in *Ctnna1* mutants (Lechler and Fuchs, 2005). It is possible that this could be due to different antibodies that were used in these studies, the fact that different models of *Ctnna1* loss were used, or potentially different ages that were examined. For example, we focused on E16.5 while it is not stated which age was examined in the Lechler & Fuchs study (Figure 4D).

Regarding delamination, in order to clarify this point, we have edited the Results section: “While we did not observe clear delamination events in our 3-6 hour live imaging experiments, genetic lineage tracing revealed that a similar and significant fraction of clones in both WT and *Afdn^2711^* epidermis (48% vs. 40%) arose from delamination (Figure 8—figure supplement 1I).” However, we have not specifically searched for delamination events in our E14.5 videos since our focus was on mitotic cells, thus we do not feel comfortable concluding whether or not they occur, or at what frequency. To our knowledge, as acknowledged by this reviewer, the only lab that has reported observing delamination by live imaging is the Wickstrom lab (Miroshnikova et al., 2018), which used LifeAct-GFP in E15.5 embryos. It is possible this discrepancy between her lab and the others mentioned could be due to differences in imaging preparations, transgenic mouse lines, and/or age. However, we agree with the

Wickstrom lab with respect to the idea that delamination does occur, particularly in early stratification. Our lab has now utilized two different lineage tracing approaches – lentiviral Cre-RFP (Williams et al., 2014) and *Krt14^CreER^* (this study) – which have identified a significant and consistent proportion of clones that arise from apparent delamination events.

Reviewer #3:This paper is improved and has addressed some of my major concerns. The lineage tracing data have helped to show that there is a likely direct consequence in terms of cell density in the different layers, even though if I understand Figure 7G and H correctly, this seems to have no direct effect on epidermal thickness or differentiation (time point of analysis is not mentioned though in the legends of these panels).

We applaud the reviewer for recognizing that the *Krt14^Cre^; Afdn^fl/fl^*cKOs do not show increased stratification compared to *Afdn^fl/fl^*WT controls (Figure 7G, H), which is at apparent odds with our later conclusion that *Afdn^2711^* and lentiviral Cre-RFP *Afdn^fl/fl^*cKOs show hyper-stratification (Figure 8—figure supplement 1D, E).However, we think this is most likely due to differences in the timing of *Afdn* targeting in these different lines. We have previously shown that lentiviral Cre (targeted to the amniotic fluid at E9.5) acts 2-3 days earlier than *Krt14^Cre^* (Beronja et al., 2010; Supplementary Figure 4). We have added a sentence to clarify this point in the Results section: “Moreover, in contrast to lentiviral-delivered *Afdn^2711^, Krt14^Cre^*-mediated deletion of *Afdn* did not cause obvious differentiation defects (compare Figure 7H to Figure 8—figure supplement 1E), likely because *Afdn* deletion occurs later with *Krt14^Cre^* (Beronja et al., 2010).”

I do still feel that the authors conclusions on tension regulation are too strong as it is all based on staining (Figures 3 and 4) and should be tuned down in the text and legend titles as no experiment directly addresses the role of tension or hyper contractility. For example, the phosphomyosin staining is suggestive of increased contractility, but there is no direct evidence for this (laser ablation, etc.) and in addition the resolution in the images does not really allow one to make this very clear statement as in the legend title that a hyper contractile basal endfoot directs planar telophase correction. I realise that such experiments, especially in vivo are not trivial, but even in none of their in vitro experiments the authors directly test tension, but it is all based on staining. The authors should not misunderstand me, but the strength is the telophase correction pathway and the role of junctions in this pathway that is independent of LGN, but whether these are direct consequences of tension is beyond this manuscript and can be suggested and discussed but requires quite a bit more to show that convincingly.Having said that, and if the authors are willing to tune those conclusions down, I think this is a very nice contribution to eLife.

We have tempered our language regarding tension regulation throughout the text. Some examples are:

– Results header: “Telophase corrective basal contacts are hypercontractile” changed to “Telophase corrective basal contacts display hallmarks of elevated actomyosin contractility”

– Results: “This anisotropy suggests that the basal endfoot is hypercontractile during telophase correction and that this contractility may function to pull the apical daughter back into the basal layer.” changed to “This anisotropy suggests that the basal endfoot may be enriched in contractile actomyosin, which we speculate may serve the function of pulling the apical daughter back into the basal layer.”

– Results: “In agreement with the observed hypercontractility in the basal endfoot of oblique telophase cells…” changed to “In agreement with the observed increase in pMLC2 in the basal endfoot of oblique telophase cells…”

– Results: “These data suggest that increased actomyosin contractility and AJ-tension may play a significant role in planar directed telophase correction.” changed to “These data suggest that increased actomyosin contractility and associated conformational changes to α-E-catenin could play a role in planar directed telophase correction.”

– Results: “These data demonstrate that α-E-catenin and vinculin are required for the proper maturation of tensile AJs.” changed to “These data demonstrate that α-E-catenin and vinculin are required for the proper maturation of AJs.”

– Results: “Collectively, these data suggest that afadin is a novel regulator of AJ tension sensitivity by affecting α-E-catenin conformation and vinculin recruitment.” changed to “Collectively, these data suggest that afadin is a novel regulator of AJ maturation by affecting α-E-catenin conformation and vinculin recruitment.”

– Discussion: “…where the direction of correction is dependent on contact with the basement membrane via a hypercontractile basal endfoot.” changed to “…where the direction of correction is dependent on contact with the basement membrane via a basal endfoot.”

– Figure 3 Legend title: “Maintenance of basal contact through a tensile endfoot correlates with planar-directed telophase correction” changed to “Maintenance of basal contact correlates with planar-directed telophase correction”

– Figure 3—figure supplement 1 Legend title: “A hypercontractile basal endfoot directs planar telophase correction in WT and LGN knockdown samples” changed to “A basal endfoot mediates planar telophase correction”

– Figure 4 Legend title: “Vinculin, α-E-catenin and afadin regulate tension and AJ linkage to the actin cytoskeleton” changed to “Vinculin and afadin regulate α-E-catenin conformation and AJ linkage to the actin cytoskeleton”